# Sampling from Structured Log-Concave Distributions via a Soft-Threshold Dikin Walk

**Oren Mangoubi**
Worcester Polytechnic Institute

**Nisheeth K. Vishnoi**
Yale University

## Abstract

Given a Lipschitz or smooth convex function $f : K \to \mathbb{R}^d$ for a bounded polytope $K := \{\theta \in \mathbb{R}^d : A\theta \leq b\}$, where $A \in \mathbb{R}^{m \times d}$ and $b \in \mathbb{R}^m$, we consider the problem of sampling from the log-concave distribution $\pi(\theta) \propto e^{-f(\theta)}$ constrained to $K$. Interest in this problem derives from its applications to Bayesian inference and differential privacy. We present a generalization of the Dikin walk to this setting that requires at most $O((md + dL^2R^2) \times md^{\omega-1}\log(\frac{w}{\delta}))$ arithmetic operations to sample from $\pi$ within error $\delta > 0$ in the total variation distance from a $w$-warm start. Here $L$ is the Lipschitz constant of $f$, $K$ is contained in a ball of radius $R$ and contains a ball of smaller radius $r$, and $\omega \approx 2.37$ is the matrix-multiplication constant. This improves on the running time of prior works for a range of structured settings important for the aforementioned inference and privacy applications. Technically, we depart from previous Dikin walks by adding a soft-threshold regularizer derived from the Lipschitz or smoothness properties of $f$ to a barrier function for $K$ that allows our version of the Dikin walk to propose updates that have a high Metropolis acceptance ratio for $f$, while at the same time remaining inside the polytope $K$.

## 1 Introduction

We consider the problem of sampling from a log-concave distribution supported on a polytope: Given a polytope $K := \{\theta \in \mathbb{R}^d : A\theta \leq b\}$, where $A \in \mathbb{R}^{m \times d}$ and $b \in \mathbb{R}^m$, and a convex function $f : K \to \mathbb{R}$, output a sample $\theta \in K$ from the distribution $\pi(\theta) \propto e^{-f(\theta)}$. Our interest in this problem derives from its applications to Bayesian inference and differentially private optimization. In Bayesian inference, the ability to sample from $\pi(\theta) \propto e^{-f(\theta)}$ allows one to compute Bayesian confidence intervals and other statistics for the Bayesian posterior distribution of many machine learning models (see e.g. [17, 20, 6]). In differentially private optimization, sampling from the "exponential mechanism" [36] allows one to get optimal utility bounds for the problem of minimizing $f$ under $\varepsilon$-differential privacy [2].

The instances of the polytope-constrained sampling problem that arise in these applications are more general than the two well-studied special cases– the uniform density case ($f \equiv 0$) and the unconstrained case ($K = \mathbb{R}^d$). However, they still have more *structure* than the case of a general log-concave function supported on an arbitrary convex body. For instance, in Bayesian Lasso logistic regression, $f(\theta) = \sum_{i=1}^n \ell(\theta; x_i)$, where $\ell$ is the logistic loss and $x_i$ are datapoints with $\|x_i\|_2 \leq 1$, and $K = \{\theta \in \mathbb{R}^d : \|\theta\|_1 \leq O(1)\}$; see [44, 43, 26, 47]. Since the logistic function is both $O(1)$-smooth and $O(1)$-Lipschitz, $f$ is both $O(n)$-Lipschitz and $O(n)$-smooth, and $K$ is defined by $O(d)$ inequalities and contained in a ball of radius $O(1)$.

To obtain an $\varepsilon$-differentially private mechanism for the Lasso logistic regression problem, using the exponential mechanism, the goal is to sample from $\exp(-\frac{\varepsilon}{R}\sum_{i=1}^n \ell(\theta; x_i))$, where $\ell$ is the logistic loss and $K$ is contained in a ball of radius $R$. Thus, the log-density is both

37th Conference on Neural Information Processing Systems (NeurIPS 2023).

$\beta$-smooth and $L$-Lipschitz for $\beta = L = {n\varepsilon}/{R} = O(d)$ if $n = d$ and $\varepsilon < 1$, since $R = O(1)$. Another example is a result of [28] that reduces the problem of $\varepsilon$-differentially private low-rank approximation of a symmetric $p \times p$ matrix to a constrained sampling problem where $f$ is linear (and thus 0-smooth) of dimension $d = p^2$ and $K$ is the Gelfand-Tsetlin polytope (a generalization of the probability simplex). Here $K$ has $d$ inequalities and diameter $O(\sqrt{d})$.

Importantly, when sampling from the exponential mechanism in privacy applications, sampling with total variation (TV) bounds–the case that has received the most attention–is insufficient to guarantee $\varepsilon$-differential privacy, the strongest notion of differential privacy; see [14]. Instead, one requires bounds in the stronger infinity-distance metric $\mathrm{d}_\infty(\nu, \pi) := \sup_{\theta \in K} |\log(\nu(\theta)/\pi(\theta))|$. A recent work [35] showed how to convert samples within $O(\delta)$-TV distance from continuous log-Lipschitz densities $\pi$, into samples with $O(\varepsilon)$-infinity-distance bounds, but it requires the TV distance $\delta$ to be *very small*–roughly $\delta = O(\varepsilon e^{-d - LR})$, raising the question of designing Markov chains whose runtime bounds also have a low-order dependence on $\log 1/\delta$. Thus, for the aforementioned applications to Bayesian inference and privacy which give rise to structured instances of sampling from log-concave densities over polytopes, it is desirable to design sampling algorithms that have a low-order polynomial dependence not only on the parameters $d, L, R, \beta$, but also on $\log 1/\delta$.

**Main related works.** A line of work has developed algorithms for sampling in the general setting when $K$ is an arbitrary convex body given by a membership oracle [16, 1, 15, 33, 34]. [33] use the "hit-and-run" framework to give an algorithm to sample from a log-concave distribution $\pi \propto e^{-f}$ on a convex body $K$ which also contains a ball of radius $r$ with TV-error $\delta > 0$ in $O(d^2(R/r)^2 \log^2(wdR/(r\delta)) \log^3(w/\delta))(T_f + \hat{T}_K))$ arithmetic operations from a $w$-warm start (their Theorem 1.1) and $O(d^3(R/r)^2) \log^5(dR^2/(\delta r)(T_f + \hat{T}_K))$ arithmetic operations from a cold start (Corollary 1.2). Here $T_f$ is the time required to evaluate $f$ and $\hat{T}_K$ is the time for a membership oracle query. Here, a distribution $\nu$ is $w$-warm for $w \geq 1$ w.r.t. the stationary distribution $\pi$ if $\sup_{z \in K} \nu(z)/\pi(z) \leq w$.

[38] give a "Dikin-walk"-based algorithm to sample from any log-concave $\pi \propto e^{-f}$ on $K$ where $f$ is $L$-Lipschitz or $\beta$-smooth. The Dikin walk Markov chain was introduced in [24] in the special case where $f \equiv 0$ (see also [37]). Their runtime is $O((d^5 + d^3 L^2 R^2) \log(w/\delta)(T_f + md^{\omega-1}))$ arithmetic operations, where $\omega = 2.37\cdots$ is the matrix-multiplication constant. From a cold start, their runtime is $O((d^5 + d^3 L^2 R^2) \log(1/\delta) (d \log(R/r) + M + \log(1/\delta))(T_f + md^{\omega-1})$, where $M := \log(\max_{\theta \in K} e^{f(a) - f(\theta)})$. Their bounds when $f$ is $\beta$-smooth are the same, but with each $L^2$ term replaced with $\beta$. We discuss additional related work in Appendix A.

**Our contributions.** We present a Markov chain sampling algorithm that generates samples from an $L$-log-Lipschitz or $\beta$-log-smooth log-concave distribution $\pi \propto e^{-f}$ on an $R$-bounded polytope $K$ given by $m$ inequalities, with an error bounded in the TV distance (Algorithm 1 and Theorem 2.1). Our algorithm requires $O((md + dL^2R^2) \log(w/\delta))(T_f + md^{\omega-1})$ arithmetic operations to sample with TV error $O(\delta)$ from $\pi \propto e^{-f}$ when $f$ is $L$-Lipschitz, and $O((md + d\beta R^2) \log(w/\delta))(T_f + md^{\omega-1})$ arithmetic operations in the setting where $f$ is $\beta$-smooth, where $T_f$ is the number of arithmetic operations to compute the value of $f$ and $md^{\omega-1}$ is the number of arithmetic operations to compute the Hessian of the log-barrier of $K$.

In comparison to [33], we improve dependence of the running time on the parameters $R/r$ and $\log 1/\delta$. In comparison to [38], we improve the dependence on $d$ while retaining the same dependence on $1/\delta$. Our result directly implies faster runtimes with improved dependence on the dimension $d$ for structured inference problems such as Bayesian Lasso logistic regression where e.g. $R/r = \Omega(\sqrt{d})$ (Corollary B.1). Moreover, plugging our algorithm into the TV-to-infinity distance bound converter of [35], we obtain an algorithm to sample from a logconcave density constrained to a polytope with error bounded by infinity distance that improves upon prior work of [2]; see Corollary C.1 and the subsequent discussion. Corollary C.1, along with the exponential mechanism [36], allows us to obtain faster runtime bounds with improved

dependence on $d$ for applications to differentially private empirical risk minimization [2] and matrix approximation [28]; see Corollary D.1 and the subsequent discussion.

Technically, our algorithm is a Markov chain inspired by the Dikin walk [24, 37], whose steps are determined by a barrier function that generalizes the log-barrier function by adding a "soft-threshold" $\ell_2$-norm regularizer. The regularized barrier allows our Markov chain to take larger steps, while still retaining a high acceptance probability on Lipschitz or smooth log-densities $f$ –allowing our Markov chain to sample from these distributions with a faster runtime. A key technical step in obtaining our results is to show that our self-concordant barrier function is the limit of an infinite sequence of log-barrier functions for $K$ "padded" with additional redundant inequalities. This allows us to leverage well-known properties of the log-barrier to bound the acceptance probability and mixing time of our Markov chain.

While $\ell_2$ regularization is optimal for classes of functions $f$ which are Lipschitz or smooth in the $\ell_2$-norm, for other classes of functions (e.g., Lipschitz in the $\ell_1$-norm), $\ell_2$ regularization may not be optimal. Moreover, it remains open to obtain runtime bounds for the Dikin walk that do not require $f$ to be Lipschitz or smooth, and/or depend polynomially on $\log R$. This leads to the question of whether one can design other tractable self-concordant barriers to obtain further runtime improvements for sampling log-concave distributions on a polytope; we discuss this in Appendix F.

## 2 Results

Our main result (Theorem 2.1 and Algorithm 1) is a Markov chain algorithm that generates TV-error bounded samples from $L$-log-Lipschitz or $\beta$-log-smooth log-concave distributions on a polytope. As explained later in this section, Theorem 2.1 often results in the fastest known algorithm for some of the applications to Bayesian inference and differentially private optimization mentioned in the introduction.

**Notation.** In the following, $f$ is $L$-Lipschitz or $\beta$-smooth for some $L, \beta > 0$. $T_f$ denotes the number of arithmetic operations to evaluate $f$, $T_K$ is the number of arithmetic operations to compute the Hessian of the log-barrier function, $\hat{T}_K$ the operations for a membership oracle query, and $\tilde{T}_K$ the operations to compute a projection oracle for $K$. When $K$ is a polytope $K = \{\theta \in \mathbb{R}^d : A\theta \leq b\}$ given by $A \in \mathbb{R}^{m \times d}$ and $b \in \mathbb{R}^m$, one has $T_K = O(md^{\omega-1})$, $\hat{T}_K = O(md)$, and $\tilde{T}_K = O(md^{\omega-1})$. For every $j \in \{1, \ldots, m\}$, we denote the $j$'th row of $A$ by $a_j$ and the $j$'th entry of $b$ by $b_j$. When comparing runtimes, we often assume for simplicity that $T_f = \Theta(d^2)$ unless otherwise stated, which is the case, e.g., in logistic regression with $n = \Theta(d)$ datapoints. For any two distributions $\mu, \nu$ on $\mathbb{R}^d$, we denote their total variation distance by $\|\mu - \nu\|_{\text{TV}} := \sup_{S \subseteq \mathbb{R}^d} |\mu(S) - \nu(S)|$. For $\theta \in \mathbb{R}^d$, $t > 0$, denote the ball of radius $t$ at $\theta$ by $B(\theta, t) := \{z \in \mathbb{R}^d : \|z - \theta\|_2 \leq t\}$ where $\|\cdot\|_2$ is the Euclidean norm. Denote the interior of any $S \subseteq \mathbb{R}^d$ by $\text{Int}(S) := \{\theta \in S : B(\theta, t) \subseteq S \text{ for some } t > 0\}$. $\alpha > 0$ is a step-size hyperparameter shared by our algorithm and the original Dikin walk of [24, 37], $\gamma > 0$ is the hyperparameter in [38], and $\eta > 0$ a hyperparameter for our algorithm's regularizer.

**Theorem 2.1 (Sampling with TV bounds via a soft-threshold Dikin Walk)**
*There exists an algorithm (Algorithm 1) which, given $\delta, R > 0$ and either $L > 0$ or $\beta > 0$, $A \in \mathbb{R}^{m \times d}$, $b \in \mathbb{R}^m$ that define a polytope $K := \{\theta \in \mathbb{R}^d : A\theta \leq b\}$ such that $K$ is contained in a ball of radius $R$ and has nonempty interior, an oracle for the value of a convex function $f : K \to \mathbb{R}^d$, where $f$ is either $L$-Lipschitz or $\beta$-smooth, and an initial point sampled from a distribution supported on $K$ which is $w$-warm with respect to $\pi \propto e^{-f}$ for some $w > 0$, outputs a point from a distribution $\mu$ where $\|\mu - \pi\|_{\text{TV}} \leq \delta$. Moreover, this algorithm takes $O((md + dL^2R^2)\log(w/\delta)) \times (T_f + T_K)$ arithmetic operations in the setting where $f$ is $L$-Lipschitz, or $O((md + d\beta R^2)\log(w/\delta)) \times (T_f + T_K)$ arithmetic operations when $f$ is $\beta$-smooth, where $T_f$ is the number of operations to evaluate $f$ and $T_K = O(md^{\omega-1})$.*

Theorem 2.1 improves on the previous bound of ([33]; Theorem 1.1) of $O(d^2(R/r)^2 \log^2(wdR/(r\delta)) \log^3(w/\delta))(T_f + \hat{T}_K)$ arithmetic operations, for sampling from $\propto e^{-f}$

with TV error $O(\delta)$ from a $w$-warm start, by a factor of $\min(d^{2-\omega}(R/r)^2, d^{3-\omega}/(r^2L^2))\log^4(w/\delta)$ when $f$ is $L$-Lipschitz on a polytope $K$ defined by $m = O(d)$ inequalities (see Table 1). Thus, $\hat{T}_K = O(md)$ and if, e.g., we also have $LR = O(\sqrt{d})$ and $R/r = \sqrt{d}$, the improvement is $d^{3-\omega}$ arithmetic operations.[1] When $f$ is $\beta$-smooth, the improvement is $\min(d^{2-\omega}(R/r)^2, d^{3-\omega}/(r^2\beta))\log^4(w/\delta)$ arithmetic operations.

When a warm start is not provided, Algorithm 1 takes at most $O((md+dL^2R^2)\times(d\log(R/r)+M+\log(1/\delta)))(T_f+T_k)$ arithmetic operations when $f$ is $L$-Lipschitz (or $O((md+d\beta R^2)\times(d\log(R/r)+M+\log(1/\delta)))(T_f+T_k)$ arithmetic operations when $f$ is $\beta$-smooth), since an $e^{d\log(R/r)+M}$-warm start can be obtained by sampling uniformly from the ball $B(a,r)\subseteq K$, where $M = \log(\max_{\theta\in K}e^{f(a)-f(\theta)}) \leq LR$. In comparison, the work of [33] for the hit-and-run algorithm gives a bound of $O(d^3(R/r)^2)\log^5(dR^2/(\delta r))(T_f+\hat{T}_K)$ arithmetic operations to sample with TV error $O(\delta)$ without a warm start. Thus, when we are not given a warm start and $\pi \propto e^{-f}$ is constrained to a polytope $K$ defined by $m = O(d)$ inequalities, Theorem 2.1 improves on the bounds of [33] for the hit-and-run algorithm by a factor of $\min(d^{2-\omega}(R/r)^2, d^{3-\omega}/(rL^2))\log^4(1/\delta)$ arithmetic operations in the setting when $f$ is $L$-Lipschitz, and $\min(d^{2-\omega}(R/r)^2, d^{3-\omega}/(r\beta))\log^4(1/\delta)$ when $f$ is $\beta$-smooth. On the other hand, we note that the results of [33] apply more generally when $\pi$ is a log-concave distribution on a convex body, while our bounds for the soft-threshold Dikin walk (Theorem 2.1) apply to the setting where $\pi$ is a log-Lipschitz (or log-smooth) log-concave distribution on a polytope. In the example of Bayesian Lasso logistic regression, our algorithm takes $O(d^{3+\omega}\log(d/\delta))$ arithmetic operations without a warm start since $f$ is both $\beta$-smooth and $L$-Lipschitz, with $\beta = L = n = m = d$, and $R = 1$ and $r = 1/\sqrt{d}$ (Corollary B.1 in Appendix B); this improves by a factor of $d^{3-\omega}\log^4(d/\delta)$ arithmetic operations on the bound of $d^6\log^5(d/\delta)$ *arithmetic operations* for the hit-and-run algorithm of [33]. This is because, while it only takes $\hat{T}_K = O(d)$ arithmetic operations to establish membership in the $\ell_1$ ball, computing the function $f$ in logistic regression requires $T_f = nd = d^2$ arithmetic operations.

| Algorithm | Iterations for $L$-Lipschitz $f$ | Iterations for $\beta$-smooth $f$ | Arithmetic operations per iteration | Iterations if $m = O(d), L, r = O(1)$, $R = O(d)$ | Iterations if $m = O(d), r = O(\frac{1}{\sqrt{d}})$, $\beta = L = O(d), R = O(1)$ |
|---|---|---|---|---|---|
| Proximal Langevin MC [3] | $\tilde{O}(d^5\delta^{-6}L^2(\frac{R}{r})^4)$ | — | $O(md^{\omega-1})+T_{\nabla f}$ | $\tilde{O}(d^9\delta^{-6})$ | $\tilde{O}(d^9\delta^{-6})$ |
| Hit-and-run [32] | $\tilde{O}(d^2(\frac{R}{r})^2)$ | same | $O(md)+T_f$ | $\tilde{O}(d^4)$ | $\tilde{O}(d^3)$ |
| Dikin Walk of [37] | $\tilde{O}(d^5+d^3L^2R^2)$ | $\tilde{O}(d^5+d^3\beta R^2)$ | $O(md^{\omega-1})+T_f$ | $\tilde{O}(d^5)$ | $\tilde{O}(d^5)$ |
| **Soft Threshold Dikin Walk (this paper)** | $\tilde{O}(md+dL^2R^2)$ | $\tilde{O}(md+d\beta R^2)$ | $O(md^{\omega-1})+T_f$ | $\tilde{O}(d^3)$ | $\tilde{O}(d^2)$ |

Table 1: Number of iterations (and arithmetic operations per-iteration) of different algorithms which imply bounds for sampling within TV error $O(\delta)$ from a logconcave $\pi \propto e^{-f}$ on a polytope $K$ when $f$ is L-Lipschitz or $\beta$-smooth, from a $w$-warm start. $T_f$ and $T_{\nabla f}$ are, respectively, the number of operations to compute $f$ or $\nabla f$. The $\tilde{O}$ notation hides logarithmic factors of $d, \delta, r, R, w$. The fifth column gives runtimes when $K$ is a polytope with $R = O(d)$ that contains the unit ball (and is thus not well-rounded), and $f$ is $O(1)$-Lipschitz. The sixth column corresponds to sampling a Bayesian Lasso logistic regression posterior distribution with $O(d)$ datapoints, where $K$ is the unit $\ell_1$-ball.

We also compare to the bound of $O((d^5+d^3L^2R^2)\log(w/\delta))(T_f+T_k)$ arithmetic operations to sample from an $L$-log-Lipschitz density on a polytope defined by $m$ inequalities with TV error $O(\delta)$ from a $w$-warm start of [38].[2] We note that while this bound has a larger dependence on $d$ (and $L$) than the bounds for the hit-and-run algorithm, the dependence on $\log 1/\delta$ is much smaller– by a factor of $\log^4 1/\delta$– which can allow for faster runtimes when the goal is to sample with infinity-distance error. We explain this in detail below. Theorem 2.1 improves on the bound of [38] bound by a factor of $d^2$ arithmetic operations if $m = O(d)$,

---

[1]Any convex body $K$ contained in a ball of radius $R > 0$ and containing a ball of smaller radius $r > 0$ satisfies the lower bound $R/r \geq 1$. Moreover, for any convex body $K$, there always exists a linear transformation $T$ for which $TK$ is contained in a ball of radius $\hat{R}$ and contains a ball of radius $\hat{r}$ such that $\hat{R}/\hat{r} \leq O(\sqrt{d})$ ($TK$ is referred to as a "well-rounded" convex body).

[2]When setting their scalar step size hyperparameter $\alpha^{1/2}$ to $O(\min(1/d, 1/LR))$, [38] get a bound of $\phi \geq \alpha^{1/2}/(\kappa\sqrt{d})$ on the conductance $\phi$ of their Dikin walk (their Lemma 4). Here $\kappa$ is the self-concordance parameter of the barrier function; for the log-barrier $\kappa = m$, although there are other barriers for which $\kappa = O(d)$. Plugging their conductance bound into Corollary 1.5 of [31] implies a bound of $\phi^{-2}\log(w/\delta)(T_f+T_k) = O((d^5+d^3L^2R^2)\log(w/\delta)(T_f+T_k))$ arithmetic operations from a $w$-warm start for their walk to sample with $O(\delta)$ TV error from $\pi$. See Appendix A for details.

and $d^3$ if we also have that $LR = O(\sqrt{d})$. When $f$ is instead $\beta$-smooth their bound is $O((d^5 + d^3\beta R^2)\log(w/\delta))(T_f + T_k)$, and our improvement on this bound is $d^2$ arithmetic operations if $m = O(d)$, and $d^3$ if we also have that $\beta R^2 = O(d)$.

We also note that while many works, e.g. [24, 37, 27, 8], give faster bounds for the Dikin walk and its variants than the bounds in [38], these only apply in the special case when $\pi$ is the uniform distribution on $K$. The proof of Theorem 2.1 appears in Appendix E. We give an overview of the main ideas in the proof of Theorem 2.1 in Section 3. In Appendix F, we give an axiomatic approach to arrive at our barrier function and discuss possible extensions.

**Infinity-distance sampling.** In applications of sampling to differentially private optimization [36, 23, 2, 18, 28], bounds in the total variation (TV) distance are insufficient to guarantee "pure" $\varepsilon$-differential privacy, and one instead requires bounds in the infinity-distance $\mathrm{d}_\infty(\nu, \pi) := \sup_{\theta \in K} |\log(\nu(\theta)/\pi(\theta))|$; see e.g. [14]. [35] give an algorithm that converts samples from TV bounds to those bounded in infinity-distance. Namely, given $\varepsilon > 0$ and a sample from a distribution $\mu$ within TV distance $\delta \leq O(\varepsilon(R(d\log(R/r)+LR)^2/\varepsilon r)^{-d}e^{-LR})$ of $\pi$, this post-processing algorithm outputs a sample within infinity-distance $O(\varepsilon)$ from $\pi$. Plugging the TV bounds from our Theorem 2.1 into their Theorem 2.2 gives a faster algorithm to sample from a log-concave and log-Lipschitz (or log-smooth) distribution constrained to a polytope $K$, with $O(\varepsilon)$ error in $\mathrm{d}_\infty$ (Corollary C.1 in Appendix C). In particular, Corollary C.1 gives a bound of $O((md + dL^2R^2) \times [LR + d\log(Rd+LRd/r\varepsilon)])(T_f + T_K)$ arithmetic operations to sample within $O(\varepsilon)$ error in $\mathrm{d}_\infty$ from a log-concave distribution $\propto e^{-f}$ constrained to a polytope $K$ from a cold start when $f$ is $L$-Lipschitz (when $f$ is also $\beta$-smooth, the bound is $O((md + d\beta R^2) \times [LR + d\log(Rd+LRd/r\varepsilon)])(T_f + T_K))$. This improves on the bound of $O((m^2d^3 + m^2dL^2R^2) \times [LR + d\log(Rd+LRd/r\varepsilon)])(T_f + T_K)$ arithmetic operations in Theorem 2.1 of [35] by a factor of $d^3$. Moreover, it further improves on the bound of $O((md^9 + md^5L^4R^4)(1/\varepsilon^2) \times \mathrm{polylog}(1/\varepsilon, 1/r, R, L, d))(T_f + \tilde{T}_K)$ operations implied by [2]. Corollary C.1 improves on this bound by a factor of $d^{9-\omega}/\varepsilon^2$ when, e.g, each function evaluation takes $T_f = O(d^2)$ operations and $m = O(d)$ as may be the case in privacy applications.

**Differentially private optimization.** A randomized mechanism $h : \mathcal{D}^n \to \mathcal{R}$ is $\varepsilon$-differentially private ($\varepsilon$-DP) if for any datasets $x, x' \in \mathcal{D}$ which differ by a single datapoint, and any $S \subseteq \mathcal{R}$, we have $\mathbb{P}(h(x) \in S) \leq e^\varepsilon \mathbb{P}(h(x') \in S)$; see [14]. $\varepsilon$-differential privacy is the strongest notion of differential privacy, holds several advantages over weaker notions of differential privacy (see Remark 2.2), and has been widely studied in the literature [14, 25, 2].

In the application of the exponential mechanism to $\varepsilon$-DP low-rank approximation of a $p \times p$ symmetric matrix $M$ [28] (see also [19]), one wishes to sample within infinity distance $O(\varepsilon)$ from a log-linear distribution $\propto e^{-\hat{f}}$ on the Gelfand-Tsetlin polytope $K \subseteq \mathbb{R}^d$ (which generalizes the probability simplex), where $d = p^2$, and where $K$ has $m = d$ inequalities with diameter $R = O(\sqrt{d})$. In this application, the log-linear density $f$ is (trivially) 0-smooth and $d^2\sigma_1$-Lipschitz, where $\sigma_1 := \|M\|_2$ is the spectral norm of $M$. Thus, when applied to the mechanism of [28], our algorithm takes $d^{4.5+\omega}\sigma_1 \log(1/\varepsilon)$ arithmetic operations. This improves by a factor of $d^3$ on the bound of $O(d^{7.5+\omega}\sigma_1)$ arithmetic operations implied by [35, 38], and improves by a factor of $d^{11.5}\sigma_1^3/\varepsilon^2$ on the bound of $O(d^{16}\sigma_1^4/\varepsilon^2)$ implied by [2].

Consider the problem of finding an (approximate) minimum $\hat{\theta}$ of an empirical risk function $f : K \times \mathcal{D}^n \to \mathbb{R}$ under the constraint that $\hat{\theta}$ is $\varepsilon$-differentially private, where $f(\theta, x) := \sum_{i=1}^n \ell_i(\theta, x_i)$. We assume that the $\ell_i(\cdot, x)$ are $\hat{L}$-Lipschitz for all $x \in \mathcal{D}^n$, $i \in \mathbb{N}$, for some given $\hat{L} > 0$. In this setting, [2] show the minimum ERM utility bound under the constraint that $\hat{\theta}$ is pure $\varepsilon$-DP, $\mathbb{E}_{\hat{\theta}}[f(\hat{\theta}, x)] - \min_{\theta \in K} f(\theta, x) = \Theta(d\hat{L}R/\varepsilon)$, is achieved if one samples $\hat{\theta}$ from the exponential mechanism $\pi \propto \exp(-\frac{\varepsilon}{2\hat{L}R}f)$ with infinity-distance error at most $O(\varepsilon)$. Plugging Corollary C.1 into the exponential mechanism, we obtain a faster algorithm for a pure $\varepsilon$-DP mechanism which achieves the minimum expected risk (Corollary D.1). Specifically, the runtime bound implied by Corollary D.1 is $O((md + dn^2\varepsilon^2) \times (\varepsilon n + d\log(nRd/r\varepsilon))(T_f + T_K))$ arithmetic operations if each $\ell_i$ is $\hat{L}$-Lipschitz (or $O((md+dn\frac{\hat{\beta}}{\hat{L}}R\varepsilon) \times (\varepsilon n + d\log(nRd/r\varepsilon))(T_f + T_K))$ if $f$ is also $\beta$-Lipschitz). This improves upon the bound of

$O((d^{10}(1/\varepsilon^2) + \varepsilon^2 n^4 d^6) \times \text{polylog}(nRd/r\varepsilon)))(T_f + \tilde{T}_K)$ arithmetic operations in [2] by a factor of $\max(d^{10-\omega}/(\varepsilon^2 m), nd^5(1/\varepsilon))$, when the $\ell_i$ are $\hat{L}$-Lipschitz on a polytope $K$ and $f$ can be evaluated in $T_f = O(nd)$ operations. And it improves by a factor of (at least) $md$ on the bound of $O((m^2 d^3 + m^2 dn^2 \varepsilon^2)(\varepsilon n + d)\log^2(nRd/(r\varepsilon))) \times md^{\omega-1})$ operations obtained in [35].

For instance, when applying the exponential mechanism to Lasso logistic regression, if e.g. $n = d$ and $\varepsilon < 1$, our algorithm requires $O(d^{3+\omega})$ arithmetic operations, improving by $d^3$ on the bound of $d^{6+\omega}$ operations implied by [35] and by roughly $d^{9-\omega}$ on the bound of $O(d^{12})$ operations implied by [2]. In another example, when training a support vector machine with hinge loss and Lasso constraints under $\varepsilon$-DP, our algorithm requires $O(d^{4+\omega})$ arithmetic operations, improving by a factor of $d^2$ on the bound of $d^{6+\omega}$ operations implied by [35] and by $d^{8-\omega}$ on the bound of $O(d^{12})$ implied by [2]. See Appendix D for details.

**Remark 2.2 (Weaker notions of differential privacy)** *$\varepsilon$-DP holds several practical advantages over weaker notions of differential privacy (DP), including $(\varepsilon, \delta)$-DP– a notion of differential privacy where the privacy of the mechanism is allowed to fail with probability $O(\delta)$. E.g., when group privacy— privacy of subsets of $k$ individuals— must be preserved, any pure $\varepsilon$-DP mechanism is also $k\varepsilon$-DP. In contrast, $(\varepsilon, \delta)$-DP only implies $(\varepsilon, ke^{(k-1)\varepsilon}\delta)$-DP for subsets of $k$ individuals– the failure probability grows exponentially with $k$.*

## 3 Overview of proof of Theorem 2.1

Suppose we are given any polytope $K = \{\theta \in \mathbb{R}^d : A\theta \leq b\}$ defined by $m$ inequalities, and a convex $f : K \to \mathbb{R}^d$ which is $L$-Lipschitz (or $\beta$-smooth) and given by an oracle which returns $f(\theta)$ at any $\theta \in K$. Our goal is to sample from $\pi \propto e^{-f}$ on $K$ within any TV error $\delta > 0$, in a number of arithmetic operations and oracle calls that has a dependence on the dimension $d$ that is a lower-order polynomial than currently available bounds for sampling from log-Lipschitz (or log-smooth) log-concave distributions, and is logarithmic in $1/\delta$.

**Extending the Dikin walk to sample from log-concave distributions on $K$.** As a first step, we begin by attempting to generate samples from $\pi$ via the (Gaussian) Dikin walk, by extending the standard analysis given in e.g. [24, 37] for the special case when $\pi$ is uniform on $K$ to the more general case where $\pi$ is a log-Lipschitz log-concave on $K$.

In the special case where $\pi$ is the uniform distribution on $K$, from any point $\theta$ in the interior of $K$, the Dikin walk proposes updates $z = \theta + \sqrt{\alpha H^{-1}(\theta)}\,\xi$ where $\xi \sim N(0, I_d)$ and $H(\theta) = \nabla^2 \varphi(\theta)$ is the Hessian of the log-barrier function $\varphi(\theta) = -\sum_{j=1}^m \log(b_j - a_j^\top \theta)$ for $K = \{\theta \in \mathbb{R}^d : A\theta \leq b\}$, and $\alpha > 0$ is a scalar hyperparameter. To ensure that the stationary distribution of the Dikin walk is the uniform distribution on $K$, if a proposed update falls in the interior of $K$, it is accepted with probability

$$\min\left(\left(\frac{\sqrt{\det(H(z))}}{\sqrt{\det(H(\theta))}}\right) e^{\|z-\theta\|_{H(\theta)}^2 - \|\theta-z\|_{H(z)}^2}, 1\right)$$

determined by the metropolis rule; otherwise, it is rejected. The use of the log-barrier is to ensure that the steps proposed by the Dikin walk remain inside the polytope $K$ w.h.p.

The hyperparameter $\alpha$ is chosen as large as possible while still ensuring the proposed steps remain in $K$ and are accepted w.h.p. On the one hand, since the covariance matrix $\alpha H^{-1}(\theta)$ of the proposed updates is proportional to $\alpha$, larger choices of $\alpha$ allow the walk to propose larger update steps. On the other hand, if $\alpha$ is too large, the proposed steps may fall outside the polytope and be rejected w.h.p. To see how to choose $\alpha$, note that for any $\theta \in \text{Int}(K)$, the Dikin ellipsoid $D_\theta = \{w : (w - \theta)^\top H^{-1}(\theta)(w - \theta) \leq 1\}$ is contained in $K$. Thus, standard Gaussian concentration inequalities which guarantee $\|\xi\|_2 = O(\sqrt{d})$ w.h.p. imply $\theta + \sqrt{\alpha H^{-1}(\theta)}\,\xi$ is in $K$ w.h.p. if $\alpha \leq O(1/d)$. Moreover, using properties of log-barrier functions, one can show the term $\det(H(z))/\det(H(\theta))$ in the acceptance ratio is also $\Omega(1)$ for $\alpha = O(1/d)$, as is done in [24, 37]. To see why, Lemma 4.3 of [45] implies that, if $H(\theta)$ is the Hessian of a log-barrier function for $K$, its log-determinant $V(\theta) = \log(\det(H(\theta)))$ satisfies

$$(\nabla V(\theta))^\top [H(\theta)]^{-1} \nabla V(\theta) \leq O(d) \qquad \forall \theta \in \text{Int}(K). \tag{1}$$

Thus, if $\alpha \leq 1/d$, the proposed update $z = \theta + \sqrt{\alpha H^{-1}(\theta)}\,\xi$ has variance $\Omega(1)$ in the direction $\nabla V(\theta)$, and (by Gaussian concentration), $(\theta - z)^\top \nabla V(\theta) \leq O(1)$ w.h.p. This implies $V(z) - V(\theta) = \log \det(H(z)) - \log \det(H(\theta)) = \Omega(1)$, and hence $\det(H(z))/\det(H(\theta)) = \Omega(1)$.

In [38], the Dikin walk is applied to the more general problem of sampling from a $L$-log-Lipschitz (or $\beta$-log-smooth) log-concave $\pi \propto e^{-f}$ on $K$ (the problem of interest in this paper). To guarantee the walk has the correct stationary distribution $\pi$, the Metropolis acceptance probability of the proposed updates $z = \theta + \sqrt{\gamma H^{-1}(\theta)}\,\xi$, where $\gamma$ is a hyperparameter, gains an additional factor $e^{-f(z)}/e^{-f(\theta)}$. To ensure this acceptance probability remains $\Omega(1)$, they modify the scalar step size $\gamma$ such that w.h.p. the walk takes steps where $f$ changes by $O(1)$. To see how to choose $\gamma$, note that since $f$ is $L$-Lipschitz, $e^{f(z)-f(\theta)} = \Omega(1)$ if the Euclidean distance $\|z - \theta\|_2$ is $O(1/L)$. This can be shown to occur w.h.p. if $\gamma = O(1/(LR)^2)$, since the fact that the Dikin ellipsoid is in $K \subseteq B(0, R)$ implies that the eigenvalues of $H(\theta)$ are all $\leq R^2$ and hence the variance $\gamma v^\top H^{-1}(\theta)v$ of the proposed step is $\leq O(1/(dL^2))$ in any given direction $v \in \mathbb{R}^d$ (where $v$ is a unit vector). Thus, it is sufficient for them to choose $\gamma = \min(1/d, 1/(LR)^2)$ to ensure the proposed step remains in $K$ and is accepted w.h.p.

On the one hand, to ensure the Markov chain proposes steps that change $f$ by an amount at most $O(1)$ for *any* $L$-Lipschitz $f$, it is necessary and sufficient to ensure that from any point $\theta \in \text{Int}(K)$, the Markov chain makes updates which fall w.h.p. inside a Euclidean ball $B(\theta, 1/L)$ of radius $1/L$ centered at $\theta$. This is because the Lipschitz condition on $f$: $\|f(\theta) - f(z)\|_2 \leq L\|\theta - z\|_2$ for all $\theta, z \in K$ holds w.r.t. the Euclidean norm $\|\cdot\|_2$. On the other hand, to ensure that the Markov chain remains inside the polytope $K$, it is sufficient to propose steps that lie inside the Dikin ellipsoid $D_\theta = \{w : (w - \theta)^\top H^{-1}(\theta)(w - \theta) \leq 1\}$ centered at $\theta$. Roughly, the scalar step size $\gamma$ is chosen such that this ellipsoid is contained inside the Euclidean ball $B(\theta, 1/L)$, as this guarantees that w.h.p. the steps proposed by the walk will both remain in $K$ and will also not change the value of $f$ by more than $O(1)$.

However, at many points $\theta$ the Dikin ellipsoid $D_\theta$ is such that the ratio of the largest to smallest eigenvalues of $H^{-1}(\theta)$ may be very large (this ratio can grow arbitrarily large as $\theta$ approaches a face of the polytope). Thus, roughly speaking, modifying the covariance matrix of the Dikin walk by a scalar constant ($\gamma H^{-1}(\theta)$) can cause the Dikin walk to propose steps whose variance in some directions is much smaller than is required for *either* of the two goals: staying inside $K$ and staying inside the ball $B(\theta, 1/L)$ defined by the Lipschitz condition on $f$. This suggests that modifying the log-barrier function for $K$ by a scalar multiple may not be the most efficient way of extending the Dikin walk to the problem of sampling from a general $L$-log-Lipschitz (or $\beta$-log-smooth) log-concave distribution on $K$, and that one may be able to obtain faster runtimes by making other modifications to the barrier function.

**A soft-threshold regularized Dikin walk.** Before we introduce our soft-threshold Dikin walk, we first note that even in the special case where $\pi$ is the uniform distribution on $K$, the analysis in [38] does not recover the bounds given in [24, 37] for this special case, as [38] use a different runtime analysis geared to time-varying distributions studied in that paper. Namely, [38] imply a bound of $O(m^2 d^3 \log(\omega/\delta))$ steps to sample from a uniform distribution on $K$, while [24, 37] show a bound of $O(md \log(\omega/\delta))$. For this reason, we first extend the analysis of the Gaussian Dikin walk given in [24, 37] for the special case of uniform $\pi$, to the more general problem of sampling from an $L$-log-Lipschitz or $\beta$-log-smooth log-concave density. The analysis in [24, 37] uses the cross-ratio distance metric. More specifically, if for any distinct $u, v \in \text{Int}(K)$ we let $p, q$ be the endpoints of the chord in $K$ passing through $u$ and $v$ such that the four points lie in the order $p, u, v, q$, the cross-ratio distance is

$$\sigma(u, v) := \frac{\|u - v\|_2 \times \|p - q\|_2}{\|p - u\|_2 \times \|v - q\|_2}. \tag{2}$$

One can show that for any $u, v \in \text{Int}(K)$, $\sigma^2(u, v) \geq (1/(m\gamma^{-1}))\|u - v\|_{\gamma^{-1}H(u)}^2$ (see e.g. [24, 37]). Thus, as the usual Dikin walk takes steps that have roughly identity covariance matrix $I_d$ with respect to the local norm $\|u\|_{\gamma^{-1}H(\theta)} := \sqrt{u^\top \gamma^{-1}H(\theta)u}$, for $\gamma = \min(1/d, (1/(LR)^2))$, the bound we would obtain on the number of steps until the Dikin walk is within TV error $\delta$ from $\pi$ is $O(\Delta^{-1} \log(\omega/\delta)) = O((md + mL^2R^2) \log(w/\delta))$ steps from a $w$-warm start.

To obtain even faster bounds, we would ideally like to allow the Dikin walk to take larger steps by choosing a larger $\gamma$, closer to the value $1/d$ that is sufficient to ensure an $\Omega(1)$ acceptance probability in the special case when $\pi$ is uniform. Unfortunately, if e.g. $LR \geq d$, reducing $\gamma$ from $1/d$ to $(1/(LR)^2)$ may be necessary to ensure the variance of the Dikin walk steps is $O(1/(dL^2))$ in every direction, and hence that the acceptance probability is $\Omega(1)$.

To get around this problem, we introduce a new variant of the Dikin walk Markov chain for sampling from any $L$-log-Lipschitz (or $\beta$-log-smooth) log-concave distributions on a polytope $K$, which generalizes the Dikin walk introduced in [24] for sampling from $\pi$ in the special case when $\pi$ is the uniform distribution on $K$. The main difference between our Dikin walk and the usual Dikin walk of [24] (and of [38]) is that our Dikin walk regularizes the Hessian $\alpha^{-1}H(\theta)$ of the log-barrier for $K$ by adding a "soft-threshold" regularization term $\eta^{-1}I_d$ proportional to the identity matrix, where $\eta$ is a hyperparameter and $\alpha$ is the same hyperparameter appearing in the original Dikin walk of [24]. Since the log-barrier Hessian $\alpha^{-1}H(\theta)$ and regularization term $\eta^{-1}I_d$ have different scalar hyperparameters $\alpha, \eta$, we can set $\alpha$ and $\eta$ independently from each other: roughly, $\alpha^{-1}$ is chosen to be the largest value such that the Dikin ellipsoid defined by the matrix $\alpha^{-1}H(\theta)$ remains inside $K$, while $\eta^{-1}$ is independently chosen to be the largest value such that, with high probability, the steps proposed by our Markov chain remain inside the ball $B(\theta, 1/L)$ defined by the Lipschitz condition on $f$. Roughly, the addition of the soft-threshold regularization term to the log-barrier Hessian allows us to reduce the variance of the proposed steps of the Dikin walk only in those directions where a choice of $\alpha = 1/d$ would cause the variance to be greater than $1/(dL^2)$ while leaving the variance in other directions unchanged.

More specifically, the steps proposed by our soft-threshold Dikin walk are Gaussian with mean 0 and covariance matrix $\Phi^{-1}(\theta) := (\alpha^{-1}H(\theta) + \eta^{-1}I_d)^{-1}$. The addition of the soft-threshold regularization term $\eta^{-1}I_d$ allows us to ensure that the largest eigenvalues of the covariance matrix $\Phi^{-1}(\theta)$ are $\leq O(1/(dL^2))$, without reducing (by more than a constant factor) the eigenvalues which were already $\leq O(1/(dL^2))$. This allows our Dikin walk to take larger steps, while still ensuring these steps are accepted w.h.p. by the Metropolis accept/reject rule for $f$. Taking larger steps allows our Dikin walk to converge more quickly to the target distribution $\pi(\theta) \propto e^{-f(\theta)}$ on $K$.

The (inverse) covariance matrix $\Phi(\theta)$ of the steps proposed by our Dikin walk is the Hessian of the function $\psi(\theta) = \alpha^{-1}\varphi(\theta) + \eta^{-1}\|\theta\|^2$ where $\varphi(\theta) = -\sum_{j=1}^{m}\log(b_j - a_j^\top\theta)$ is the log-barrier for $K$. The modified function $\psi(\theta)$ can be seen to also be a self-concordant barrier for $K$. In the special case where $\pi$ is the uniform distribution, $L = 0$ and $\eta^{-1} = 0$, and our "soft-threshold" Dikin walk recovers the original walk of [24]. Thus, our walk generalizes the original Dikin walk to the problem of sampling from a general $L$-log-Lipschitz (or $\beta$-log-smooth) log-concave distribution on a polytope.

**Bounding the number of Markov chain steps.** Setting $\eta = 1/(dL^2)$ ensures the variance $v^\top\alpha H^{-1}(\theta)v$ in any given unit-vector direction $v$ of the proposed update $z - \theta$ of our Markov chain is at most $O(1/(dL^2))$, and hence the term $e^{f(z)-f(\theta)}$ in the Metropolis acceptance rule is $\Omega(1)$ with high probability (Lemma E.5). Moreover, we also show that, if we choose $\alpha = 1/d$, the other terms in the Metropolis acceptance rule are also $\Omega(1)$ (Lemmas E.8, E.9). While the proofs of these lemmas follow roughly the same outline as in the special case of the original Dikin walk where $\pi$ is uniform (e.g., [24, 37]), our bound on the determinantal term $\det\Phi(z)/\det\Phi(\theta)$ must deal with additional challenges, which we discuss in the next subsection.

To bound the number of steps required by our Markov chain, we first bound the cross-ratio distance $\sigma(u, v)$ between any $u, v \in \text{Int}(K)$ by the local norm $\|u - v\|_{\Phi(u)}$ (Lemma E.2):

$$\sigma^2(u, v) \geq \left(\frac{1}{2m}\sum_{i=1}^{m}\frac{(a_i^\top(u-v))^2}{(a_i^\top u - b_i)^2}\right) + \frac{1}{2}\frac{\|u-v\|_2^2}{R^2} \geq \frac{1}{2m\alpha^{-1} + 2\eta^{-1}R^2}\|u-v\|_{\Phi(u)}^2. \quad (3)$$

Using (3) together with the isoperimetric inequality for the cross-ratio distance (Theorem 2.2 of [32]), we show that, if the acceptance probability of our Markov chain is $\Omega(1)$ at each step, then the number of steps for our Markov chain to obtain a sample within a TV distance of $\delta$ from $\pi$ is $O(\Delta^{-1}\log(\omega/\delta)) = O((md + dL^2R^2)\log(w/\delta))$ from an $w$-warm start,

where $\Delta = \Omega(1/(2m\alpha^{-1}+2\eta^{-1}R^2))$. In particular, if $m = O(d)$ and $LR > d$, this improves on the bound we would get for the basic Dikin walk by a factor of $d$.

**Bounding the determinantal term in the acceptance probability.** For our mixing time bound to hold, we still need to show that $\det(\Phi(z))/\det(\Phi(\theta))$ is $\Omega(1)$ w.h.p. We would ideally like to follow the general approach previous works [24, 37] use to show the term $\det(H(z))/\det(H(\theta))$ in the basic Dikin walk is $\Omega(1)$ w.h.p., which relies on the property of log-barrier functions in Inequality (1). Unfortunately, as $\Phi(\theta)$ is not the Hessian of a log-barrier function for any system of inequalities defining $K$, we cannot directly apply (1) to $\Phi(\theta)$.

To get around this problem, we show that, while $\Phi(\theta)$ is not the Hessian of a log-barrier function, it is in fact the limit of a sequence of matrices $H_i(\theta)$, $i \in \mathbb{N}$, where each matrix $H_i(\theta)$ in this sequence *is* the Hessian of a (different) log-barrier function for $K$. Specifically, for every $j \in \mathbb{N}$, we consider the matrices $A^j = [A^\top, I_d, \ldots, I_d]^\top$ where $A$ is concatenated with $(m_j-1)/d$ copies of the identity matrix $I_d$, and $m_j = m + \lfloor \alpha\eta^{-1}j^2 \rfloor d$. And we consider the vectors $b^j = (b^\top, j\mathbf{1}^\top, \ldots, j\mathbf{1}^\top)^\top$, where $b$ is concatenated with $(m_j-1)/d$ copies of the vector $j\mathbf{1}$, where $\mathbf{1} = (1, \ldots, 1)^\top \in \mathbb{R}^d$ is the all-ones vector. Then, for large enough $j$, $K = \{\theta \in \mathbb{R}^d : A^j\theta \leq b^j\}$, and the Hessian of the corresponding log-barrier functions is

$$H_j(\theta) \quad = \sum_{i=1}^{m^j} \frac{a_i^j(a_i^j)^\top}{((a_i^j)^\top\theta - b_i^j)^2} = H(\theta) + \lfloor \alpha\eta^{-1}j^2 \rfloor \sum_{i=1}^{d} \frac{e_i e_i^\top}{(e_i^\top\theta - j)^2}. \tag{4}$$

Using this fact (4), we show that, for every $\theta \in \text{int}(K)$, every $z \in \frac{1}{2}D_\theta$, and every sequence $\{z_j\}_{j=1}^\infty \subseteq \frac{1}{2}D_\theta$ such that $z_j \to z$, we have (Lemma E.7),

$$\lim_{j\to\infty} \frac{\det(H_j(z_j))}{\det(H_j(\theta))} = \frac{\det(\Phi(z))}{\det(\Phi(\theta))}. \tag{5}$$

Moreover, since each $H_j$ is the Hessian of a log-barrier function for $K$, we have that (1) does hold for $H_j$ and hence (from the work of [24, 37]) that $\det(H_j(z_j))/\det(H_j(\theta)) = \Omega(1)$ w.h.p. for all $j \in \mathbb{N}$, if we set $z_j = \theta + \alpha^{1/2}H_j^{-1/2}(\theta)\xi$, $\xi \sim N(0, I_d)$, and choose $\alpha = 1/d$. Thus, (5) implies that $\det(\Phi(z))/\det(\Phi(\theta)) = \Omega(1)$ w.h.p. as well (Lemma E.9), and hence the acceptance probability of the soft-threshold Dikin walk is $\Omega(1)$ at each step.

**Bounding the number of arithmetic operations.** Since the acceptance probability is $\Omega(1)$, from the above discussion, the number of steps for our Markov chain to obtain a sample within TV distance $\delta > 0$ from $\pi$ is $O((md + L^2R^2)\log(w/\delta))$ from a $w$-warm start.

Each time our Markov chain proposes a step $\theta + \Phi^{-1/2}(\theta)\xi$, it must first sample a Gaussian vector $\xi \sim N(0, I_d)$ which takes $O(d)$ arithmetic operations. It must then compute the log-barrier Hessian $H(\theta)$, and invert the matrix $\Phi(\theta) = \alpha^{-1}H(\theta) + \eta^{-1}I_d$.

Since $H(\theta) = C(\theta)C(\theta)^\top$, where $C(\theta)$ is a $d \times m$ matrix with columns $c_j(\theta) = a_j/(a_j^\top\theta - b_j)$ for all $j \in [m]$, we can compute $H(\theta)$ in $md^{\omega-1}$ arithmetic operations. And since $\Phi(\theta)$ is a $d \times d$ matrix, computing $\Phi^{-1/2}(\theta)$ can be accomplished in $d^\omega$ arithmetic operations by computing the singular value decomposition (SVD) of $\Phi$. Next, we compute the acceptance probability $\min\left(\frac{e^{-f(z)}\sqrt{\det(\Phi(z))}}{e^{-f(\theta)}\sqrt{\det(\Phi(\theta))}}e^{\|z-\theta\|_{\Phi(\theta)}^2 - \|\theta-z\|_{\Phi(z)}^2}, 1\right)$. The determinants can be computed in $O(d^\omega)$ arithmetic operations via the SVD. Evaluating $f(z), f(\theta)$ takes two calls to the oracle for $f$.

Thus, from a $w$-warm start, the soft-threshold Dikin walk takes at most $O((md + dL^2R^2) \times \log(w/\delta))$ Markov chain steps to obtain a sample from $\pi$ with total variation error $\delta > 0$, where each step takes $O(md^{\omega-1})$ arithmetic operations, and one function evaluation.

**Remark 3.1 (Optimality of $\ell_2$-regularization)** *For the class of functions considered in our paper, we conjecture that an $\ell_2$-regularizer that does not depend on $\theta$ is optimal. This is because we consider the class of functions which are L-Lipschitz or $\beta$-smooth with respect to the $\ell_2$-norm, and our bound on L or $\beta$ does not depend on $\theta$. Moreover, we only have access to the function $\theta$ through an oracle which returns the value of $f$ at any given point $\theta$ but does not tell us how $f$ changes at nearby points.*

---

**Algorithm 1:** Soft-threshold Dikin walk

---

**Input:** $m, d \in \mathbb{N}$, $A \in \mathbb{R}^{m \times d}$, $b \in \mathbb{R}^m$, which define $K := \{\theta \in \mathbb{R}^d : A\theta \leq b\}$.

**Input:** Oracle returning the value of a convex $f : K \to \mathbb{R}$. Initial point $\theta_0 \in \mathrm{Int}(K)$.

**1 Hyperparameters:** $\alpha > 0$, $\eta > 0$, and $T \in \mathbb{N}$.

**2** Set $\theta \leftarrow \theta_0$

**3 for** $i = 1, \ldots, T$ **do**

**4**      Sample a point $\xi \sim N(0, I_d)$

**5**      Set $H(\theta) \leftarrow \sum_{j=1}^m \frac{a_j a_j^\top}{(a_j^\top \theta - b_j)^2}$

**6**      Set $\Phi(\theta) \leftarrow \alpha^{-1} H(\theta) + \eta^{-1} I_d$

**7**      Set $z \leftarrow \theta + \Phi(\theta)^{-1/2} \xi$

**8**      **if** $z \in \mathrm{Int}(K)$ **then**

**9**          Set $H(z) \leftarrow \sum_{j=1}^m \frac{a_j a_j^\top}{(a_j^\top z - b_j)^2}$

**10**          Set $\Phi(z) \leftarrow \alpha^{-1} H(z) + \eta^{-1} I_d$

**11**          Accept $z$ with probability $\frac{1}{2} \min \left( \frac{e^{-f(z)} \sqrt{\det(\Phi(z))}}{e^{-f(\theta)} \sqrt{\det(\Phi(\theta))}} \times e^{\|z - \theta\|_{\Phi(\theta)}^2 - \|\theta - z\|_{\Phi(z)}^2}, 1 \right)$

**12**      **else**

**13**          Reject $z$

**14**      **end**

**15 end**

**16** Output $\theta$

---

In Theorem 2.1, we set the step size hyperparameters $\alpha = 1/(10^5 d)$ and $\eta = 1/(10^4 dL^2)$ if $f$ is $L$-Lipschitz, and the number of steps to be $T = 10^9 \left( 2m\alpha^{-1} + \eta^{-1} R^2 \right) \times \log(w/\delta)$. When $f$ is $\beta$-smooth (but not necessarily Lipschitz), we instead set $\alpha = 1/(10^5 d)$ and $\eta = 1/(10^4 d\beta)$.

In many applications, a bound on $L$ or $\beta$ can be calculated analytically, allowing one to set $\alpha, \eta$ as above. This includes, e.g., applications to training Bayesian or differentially private logistic regression models (or other generalized linear models such as support vector machines). When a bound on $L$ or $\beta$ is not known, one can in practice set $\alpha, \eta$ by hand such that the average acceptance probability is $\geq \Omega(1)$.

## 4 Conclusions, limitations, and future work

Our result improves on the runtime bounds of a line of previous work, for the problem of sampling from several classes of log-Lipschitz or log-smooth log-concave distributions on a polytope (see Table 1). To the best of our knowledge, this is the first result to introduce regularized barrier functions that simultaneously take into account the geometry of both the constraint polytope and the Lipschitz or smoothness property of a target logconcave function. These barrier functions may be of independent interest for sampling or optimization.

On the other hand, we note that [27] give an implementation of the Dikin walk in the special case where $f$ is constant, where the (average) cost of computing the Hessian matrix of the log-barrier of the polytope $K := \{\theta \in \mathbb{R}^d : A\theta \leq b\}$ at each step of the walk is improved to roughly $O(d^2 + \mathrm{nnz}(A))$ arithmetic operations, where $\mathrm{nnz}(A)$ denotes the number of non-zero entries of $A$. Whether this improvement in the per-step computation time can be achieved for the problem of computing the regularized barrier functions used in our algorithm, in the more general setting where $f$ is $L$-Lipschitz or $\beta$-smooth, is an interesting open problem.

Moreover, we note that our bounds are polynomial in $L$ or $\beta$, yet there are algorithms for sampling from log-concave distributions $\propto e^{-f}$ which do not assume $f$ is $L$-Lipschitz or $\beta$-smooth. Thus, another interesting open problem is whether one can obtain runtime bounds for a version of the Dikin walk which do not require $f$ to be Lipschitz or smooth.

Our results have applications to Bayesian inference and differentially private optimization. Bayesian inference can lead to algorithms with better generalization properties and quantification of uncertainty, and differential privacy guarantees are important to protecting the privacy of individuals in medical and other sensitive datasets. Thus, we believe our results will have positive societal impacts, and do not anticipate any negative impacts to society.

**Acknowledgments.** NV was supported in part by an NSF CCF-2112665 award. OM was supported in part by an NSF CCF-2104528 award and a Google Research Scholar award.

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

# Contents

# A   Additional discussion with related work

The problem of sampling from the uniform distribution when $K$ is a polytope given by a set of inequalities, and the distance to $\pi$ is measured in total-variation (TV) distance has been widely studied; see, e.g., [24, 37, 30, 8, 27]. Also widely studied is the problem of sampling from an unconstrained log-concave distribution. One direction of study includes works that require the log-density $\pi$ to be $L$-log-Lipschitz or $\beta$-log-smooth on *all* $\mathbb{R}^d$ for some $L, \beta > 0$, and give bounds on the distance to $\pi$ in terms of TV distance [13], Wasserstein distance [12, 9, 42], and Kullback-Leibler (KL) divergence [46, 11].

Further, [5, 3] provide versions of the Langevin dynamics for sampling from a log-concave distribution $\pi \propto e^{-f}$ on $K$ where $f$ is $L$-Lipschitz and $\beta$-smooth and $K$ is given by a projection oracle– when, e.g., $K$ is contained in a ball of radius $O(1)$, it contains a ball of radius $\Omega(1)$, and $L = \beta = O(1)$, [5] give a bound of roughly $O(d^9/\delta^{22})$ gradient and projection oracle calls to sample from $\pi$ with TV error $\delta > 0$, while [3] give a bound of roughly $O(d^5/\delta^6)$ gradient and projection oracle calls. [21] provide an algorithm for sampling from distributions $\propto e^{-f}$ where $f$ is both $L$-Lipschitz and $\mu$-strongly convex (that is, $f(\theta)$ is the sum of a convex function $\hat{f}(\theta)$ and the quadratic function $(\mu/2)\|\theta\|^2$ for some $\mu > 0$) on a convex body $K$ in roughly $\tilde{O}((L^2/\mu)\log^2(d/\delta))$ membership and function oracle calls (see also [7], who require $f$ to be $\mu$-strongly convex for $\mu > 0$ and $\beta$-smooth). However, the algorithms of [21, 7] do not apply to the more general setting where $f$ is convex and $L$-Lipschitz (or $\beta$-smooth), but not necessarily $\mu$-strongly convex for any $\mu > 0$.

**Detailed comparison to [38].**   Compared to [38], in our version of the Dikin walk, $z$ is sampled from a Gaussian with covariance matrix

$$(\alpha^{-1}H(z) + \eta^{-1}I_d)^{-1},$$

where $\alpha^{-1}$ and $\eta^{-1}$ are hyper-parameters chosen to be $\alpha^{-1} \approx d^2$ and $\eta^{-1} \approx dL^2$ if $f$ is $L$-Lipschitz (or $\eta^{-1} \approx d\beta$ if $f$ is $L$-smooth). The "soft-threshold" regularization term $\eta^{-1}I_d$ prevents the Markov chain from taking steps where the value of $f$ decreases by more than $O(1)$ w.h.p., ensuring that the term $e^{f(\theta)-f(z)}$ in the acceptance probability is $\Omega(1)$. The soft-threshold regularizer is chosen to be a multiple of the identity matrix $I_d$ since the Lipschitz condition on $f$ is rotationally invariant– it bounds the derivative of $f$ by the same amount $L$ in each direction (the same is true for the second derivative of $f$ if $f$ is $\beta$-smooth). This in turn allows our choice of scaling $\alpha^{-1}$ for the "Dikin ellipsoid" term $H(z)$– which is not in general rotationally invariant, and determined only by the geometry of the polytope rather than the geometry of the function $f$– to be independent of $L$ (or $\beta$). This is in contrast to the Dikin walk in [38] where the scaling parameter for $H(z)$ must depend on $L$ (or $\beta$) to ensure an $\Omega(1)$ acceptance probability, which allows our Markov chain to propose steps with a larger variance than the Dikin walk in [38] in directions which are not the largest eigenvector of $H^{-1}(z)$.

The (inverse) covariance matrix $\alpha^{-1}H(z) + \eta^{-1}I_d$ of our soft-threshold Dikin walk updates is the Hessian of the function $\psi(\theta) = \alpha^{-1}\varphi(\theta) + \eta^{-1}\|\theta\|_2^2$. This barrier function can be seen to be a Hessian of a self-concordant barrier. On the other hand, it is not the Hessian of a logarithmic-barrier function for any polytope defined by any set of inequalities. This prevents us from directly applying the analysis of the Dikin walk in the special case where $f \equiv 0$ [37]–which relies on properties of log-barrier functions– to our soft-threshold Dikin walk on Lipschitz or smooth $f$. To get around this problem, we show that, while $\psi(\theta)$ is not a log-barrier function of any polytope $K$, it is the limit of a sequence of log-barrier functions $\hat{\psi}_1, \hat{\psi}_2, \ldots$ where $\hat{\psi}_i(\theta) \to \psi(\theta)$ uniformly in $\theta$ as $i \to \infty$. See Section 3 for a detailed overview of the proof.

An open problem is to obtain runtime bounds for the Dikin walk that do not require $f$ to be Lipschitz or smooth, and/or that depend polynomially on $\log R$. This leads to the question of whether one can design other tractable self-concordant barriers to obtain further runtime improvements for sampling log-concave distributions on a polytope. We discuss possible extensions in Appendix F.

**Explanation of bounds in [38].** [38] consider a version of the Dikin walk for sampling from a log-concave distribution $\pi \propto e^{-f}$ on a polytope $K$ contained in a ball of radius $R > 0$ (and also more generally on any convex body with a self-concordant barrier function). Their version of the Dikin walk has proposed updates $z = \alpha^{\frac{1}{2}} H^{-\frac{1}{2}}(\theta)\xi$ where $\xi \sim N(0, I_d)$, and acceptance probability given by $\frac{1}{2} \times \min\left(\frac{e^{-f(z)}\sqrt{\det(H(z))}}{e^{-f(\theta)}\sqrt{\det(H(\theta))}}, 1\right)$. Here $H$ is the hessian of a barrier function for the convex body $K$, which may be the log-barrier or another barrier function. For a step size of $\alpha^{\frac{1}{2}} = O(\min(\frac{1}{d}, \frac{1}{LR}))$ (Condition 4 in Section 3.1 of their paper; note that $R$ is considered a constant in [38] and therefore does appear explicitly in their bounds)[3], they obtain a bound on the conductance $\phi$ of their version of the Dikin walk (Lemma 4 of their paper):

$$\phi \geq \frac{\alpha^{\frac{1}{2}}}{\kappa\sqrt{d}}, \tag{6}$$

where $\kappa$ is the self-concordance parameter of the barrier function; for the log-barrier $\kappa = m$, although there are other barrier functions for which $\kappa = O(d)$. Plugging Inequality (6) into Corollary 1.5 of [31] implies that their Dikin walk can obtain a sample from $\pi$ with $O(\delta)$ TV error in roughly $\phi^{-2} \log(\frac{w}{\delta}) = O((d^5 + d^3 L^2 R^2) \log(\frac{w}{\delta}))$ steps from a $w$-warm start.

The number of arithmetic operations per step is $md^{\omega-1}$ and the number of arithmetic operations to obtain a sample with TV error $O(\delta)$ from $\pi$ is $O((md^{4+\omega} + md^{2+\omega}L^2R^2) \log(\frac{w}{\delta}))$. If one also has that $K$ contains a ball of radius $r > 0$, from a cold start one can find an initial point which is $e^{d \log(\frac{R}{r})+LR}$-warm by sampling from the uniform distribution on this ball, and hence their bound is $O((md^{4+\omega} + md^{2+\omega}L^2R^2)(d \log(\frac{R}{r}) + LR + \log(\frac{1}{\delta})))$.

## B  Application to Bayesian Lasso logistic regression

The following corollary follows directly from Theorem 2.1.

**Corollary B.1** *There exists an algorithm which, given $\delta, c > 0$ and a dataset $\{x_i\}_{i=1}^n \subseteq \mathbb{R}^d$ s.t. $\|x_i\|_2 \leq 1 \ \forall \ i \in [n]$, and letting $f(\theta) = \sum_{i=1}^n \ell(\theta^\top x_i)$ where $\ell(s) = \frac{1}{1+e^{-s}}$ is the logistic loss, $K = \{\theta \in \mathbb{R}^d : \|\theta\|_1 \leq c\}$ is the Lasso constraint, and $\pi \propto e^{-f}$ has support $K$, outputs a point from a distribution $\mu$ s.t. $\|\mu - \pi\|_{\mathrm{TV}} \leq \delta$, in $O((d^2 + ndc^2)(d^{\omega+1} + nd^2) \log(\frac{d}{\delta}))$ arithmetic operations.*

## C  Infinity-distance sampling

In applications of sampling to differentially private optimization [36, 23, 2, 18, 28], bounds in the total variation (TV) distance are insufficient to guarantee "pure" differential privacy, and one instead requires bounds in the infinity-distance $\mathrm{d}_\infty(\nu, \pi) := \sup_{\theta \in K} |\log(\nu(\theta)/\pi(\theta))|$; see e.g. [14]. [35] give an algorithm that converts samples from TV bounds to those bounded in infinity-distance. Namely, given $\varepsilon > 0$ and a sample from a distribution $\mu$ within TV distance $\delta \leq O\left(\varepsilon \left(R(d \log(R/r)+LR)^2/\varepsilon r\right)^{-d} e^{-LR}\right)$ of $\pi$, this post-processing algorithm outputs a sample from a distribution $\nu$ with infinity distance $\mathrm{d}_\infty(\nu, \pi) \leq \varepsilon$ from $\pi$. Plugging the TV bounds from our Theorem 2.1 into their Theorem 2.2 gives a faster algorithm to sample from a log-concave and log-Lipschitz distribution constrained to a polytope $K$, with $O(\varepsilon)$ error in $\mathrm{d}_\infty$. In particular, Corollary C.1 improves the bound of $O((m^2 d^3 + m^2 dL^2R^2) \times [LR + d \log(Rd+LRd/(r\varepsilon))])(T_f + T_K)$ arithmetic operations in their Theorem 2.1. The proof is identical to that of how their Theorem 2.2 implies their Theorem 2.1, and we refer the reader to their paper.

**Corollary C.1 (Sampling with infinity-distance guarantees)** *There exists an algorithm which, given $\varepsilon, L, r, R > 0$, $A \in \mathbb{R}^{m \times d}$, $b \in \mathbb{R}^m$ (and possibly $\beta > 0$), that define a polytope $K := \{\theta \in \mathbb{R}^d : A\theta \leq b\}$ contained in a ball of radius $R$, a point*

---

[3]There is a typo on page 7 of [38]: "R" should be the radius of the smallest ball containing $K$, not the radius of the largest ball contained in $K$.

$a \in \mathbb{R}^d$ such that $K$ contains a ball $B(a, r)$ of smaller radius $r$, and an oracle for the value of a convex function $f : K \to \mathbb{R}^d$, where $f$ is $L$-Lipschitz (or is both $L$-Lipschitz and $\beta$-smooth), and defining $\pi$ to be the distribution $\pi \propto e^{-f}$, outputs a point from a distribution $\nu$ such that $d_\infty(\nu, \pi) < \varepsilon$. Moreover, this algorithm takes $O\left((md + dL^2R^2) \times \left[LR + d\log\left(\frac{Rd + LRd}{r\varepsilon}\right)\right]\right)(T_f + T_K)$ arithmetic operations, where $T_f$ is the number of arithmetic operations to evaluate $f$ and $T_K = O(md^{\omega-1})$. If $f$ is also $\beta$-Lipschitz, the number of operations is $O\left((md + d\beta R^2) \times \left[LR + d\log\left(\frac{Rd + LRd}{r\varepsilon}\right)\right]\right)(T_f + T_K)$.

Corollary C.1 further improves the dependence on the dimension $d$ over [2, 35]. Specifically, [2] implies a bound of $O((md^9 + md^5L^4R^4)(1/\varepsilon^2) \times \text{polylog}(1/\varepsilon, 1/r, R, L, d))(T_f + \tilde{T}_K)$ arithmetic operations, where $T_f$ is the number of operations to compute $f$ and $\tilde{T}_K$ the operations to compute a projection oracle for $K$. Corollary C.1 improves on this bound by a factor of roughly $d^{10-\omega}/(\varepsilon^2 m)$ when, e.g, each function evaluation takes $T_f = O(d^2)$ operations. For example, when $m = O(d)$, as may be the case in privacy applications, the improvement is $d^{9-\omega}/\varepsilon^2$. Moreover, it improves by $d^3$ on the bound of $O((m^2d^3 + m^2d\min(L^2R^2, \beta R^2)) \times [LR + d\log((Rd + LRd)/(r\varepsilon))](T_f + T_K)$ arithmetic operations obtained in [35].

# D   Differentially private optimization

A randomized mechanism $h : \mathcal{D}^n \to \mathcal{R}$ is said to be $\varepsilon$-differentially private ($\varepsilon$-DP) if for any datasets $x, x' \in \mathcal{D}$ which differ by a single datapoint, and any $S \subseteq \mathcal{R}$, we have that $\mathbb{P}(h(x) \in S) \leq e^\varepsilon \mathbb{P}(h(x') \in S)$; see [14]. In the application of the exponential mechanism to $\varepsilon$-DP low-rank approximation of a $p \times p$ symmetric matrix $M$ [28] (see also [19]), one wishes to sample within infinity distance $O(\varepsilon)$ from a log-linear distribution $\propto e^{-f}$ on the Gelfand-Tsetlin polytope $K \subseteq \mathbb{R}^d$ (which generalizes the probability simplex), where $d = p^2$, and where $K$ has $m = d$ inequalities with diameter $R = O(\sqrt{d})$. In this application, the log-linear density $f$ is (trivially) 0-smooth and $d^2\sigma_1$-Lipschitz, where $\sigma_1 := \|M\|_2$ is the spectral norm of $M$. Thus, when applied to the mechanism of [28], our algorithm takes $d^{4.5+\omega}\sigma_1 \log(1/\varepsilon)$ arithmetic operations. This improves by a factor of $d^3$ on the runtime bound of $O(d^{7.5+\omega}\sigma_1)$ arithmetic operations implied by [35, 38], and improves by a factor of $d^{11.5}\sigma_1^3/\varepsilon^2$ on the bound of $O\left(d^{16}\sigma_1^4/\varepsilon^2\right)$ arithmetic operations for the bound in [2].

Consider the problem of finding an (approximate) minimum $\hat{\theta}$ of an empirical risk function $f : K \times \mathcal{D}^n \to \mathbb{R}$ under the constraint that $\hat{\theta}$ is $\varepsilon$-differentially private, where $f(\theta, x) := \sum_{i=1}^n \ell_i(\theta, x_i)$. We assume that the $\ell_i(\cdot, x)$ are $\hat{L}$-Lipschitz for all $x \in \mathcal{D}^n$, $i \in \mathbb{N}$, for some given $\hat{L} > 0$. In this setting, [2] show the minimum ERM utility bound under the constraint that $\hat{\theta}$ is pure $\varepsilon$-DP, $\mathbb{E}_{\hat{\theta}}[f(\hat{\theta}, x)] - \min_{\theta \in K} f(\theta, x) = \Theta(d\hat{L}R/\varepsilon)$, is achieved if one samples $\hat{\theta}$ from the exponential mechanism $\pi \propto \exp(-\frac{\varepsilon}{2\hat{L}R}f)$ with infinity-distance error at most $O(\varepsilon)$. Plugging Corollary C.1 into the exponential mechanism, we obtain a faster algorithm for a pure $\varepsilon$-DP mechanism which achieves the minimum expected risk (Corollary D.1).

**Corollary D.1 (Differentially private empirical risk minimization)** *There exists an algorithm which, given $\varepsilon, \hat{L}, r, R > 0$, $A \in \mathbb{R}^{m \times d}$, $b \in \mathbb{R}^m$ (and possibly $\hat{\beta} > 0$) that define a polytope $K := \{\theta \in \mathbb{R}^d : A\theta \leq b\}$ contained in a ball of radius $R$ and containing a ball of smaller radius $r$, and an empirical risk function $f(\theta, x) := \sum_{i=1}^n \ell_i(\theta, x_i)$, where each $\ell_i : K \to \mathbb{R}$ is $\hat{L}$-Lipschitz (and possibly also $\hat{\beta} - smooth$), outputs a random point $\hat{\theta} \in K$ which is pure $\varepsilon$-differentially private and satisfies $\mathbb{E}_{\hat{\theta}}[f(\hat{\theta}, x)] - \min_{\theta \in K} f(\theta, x) \leq O(d\hat{L}R/\varepsilon)$. Moreover, this algorithm takes $O((md + dn^2\varepsilon^2) \times (\varepsilon n + d\log(nRd/(r\varepsilon))(T_f + T_K))$ arithmetic operations if each $\ell_i$ is $\hat{L}$-Lipschitz (or $O((md + dn\frac{\hat{\beta}}{\hat{L}}R\varepsilon) \times (\varepsilon n + d\log(nRd/(r\varepsilon))(T_f + T_K))$ if $f$ is also $\beta$-Lipschitz), where $T_f$ is the number of operations to evaluate $f$ and $T_K = O(md^{\omega-1})$.*

Corollary D.1 improves on [2, 35]. In particular, it improves upon the bound of $O((d^{10}(1/\varepsilon^2) + \varepsilon^2n^4d^6) \times \text{polylog}(nRd/r\varepsilon)))(T_f + \tilde{T}_K)$ arithmetic operations in [2] by a factor of roughly $\max(d^{10-\omega}/(\varepsilon^2 m), nd^5(1/\varepsilon))$ arithmetic operations, in the setting where the $\ell_i$ are $\hat{L}$-Lipschitz

on a polytope $K$ and $f$ can be evaluated in $T_f = O(nd)$ operations ($\tilde{T}_K$ is the number of operations to compute a projection oracle for $K$). And it improves by a factor of (at least) $md$ on the bound of $O((m^2 d^3 + m^2 dn^2 \varepsilon^2) \times (\varepsilon n + d) \log^2(nRd/(r\varepsilon))) \times md^{\omega-1})$ arithmetic operations obtained in [35]. The proof is identical to that of how their Theorem 2.2 implies their Corollary 2.4, and we refer the reader to their paper. For instance, when applying the exponential mechanism to the Lasso logistic regression problem, each loss $\ell_i$ is both $\hat{\beta}$-smooth and $\hat{L}$-Lipschitz for $\hat{\beta} = \hat{L} = 1$ and $R = O(1)$. Thus, if $n = d$ and $\varepsilon < 1$, for this problem our algorithm requires $O(d^{3+\omega})$ arithmetic operations, which improves by a factor of $d^3$ on the bound of $d^{6+\omega}$ arithmetic operations implied by Corollary 2.4 in [35] and by roughly $d^{9-\omega}$ on the bound of $O(d^{12})$ arithmetic operations implied by [2]. In another example, when training a support vector machine model with hinge loss and Lasso constraints under $\varepsilon$-differential privacy, one has that $\ell_i$ is $\hat{L}$-Lipschitz but not smooth, for $\hat{L} = 1$, and $R = O(1)$. Thus, if $n = d$ and $\varepsilon < 1$, our algorithm requires $O(d^{4+\omega})$ arithmetic operations, which improves by a factor of $d^2$ on the bound of $d^{6+\omega}$ operations implied by Corollary 2.4 in [35] and by roughly $d^{8-\omega}$ on the bound of $O(d^{12})$ implied by [2].

# E    Proof of Theorem 2.1

## E.1    Bounding the number of arithmetic operations

In the following, we assume the hyperparameters $\alpha, \eta$ satisfy $\alpha \leq \frac{1}{10^5 d} \log(\frac{1}{\hat{\delta}})$ for some choice of $\hat{\delta} > 0$, and either $\eta \leq \frac{1}{10^4 dL^2}$ (in the setting where $f$ is $L$-Lipschitz) or $\eta \leq \frac{1}{10^4 d\beta}$ (in the setting where $f$ is $\beta$-smooth). To obtain our main result (Theorem 2.1), we set $\hat{\delta} = \frac{1}{100}$. To obtain improved runtime bounds in the setting where $A$ is sparse, we will set $\hat{\delta} = \frac{1}{100(md+dL^2 R^2)\log(\frac{w}{\delta})}\delta$ when $f$ is $L$-Lipschitz, and $\hat{\delta} = \frac{1}{100(md+d\beta R^2)\log(\frac{w}{\delta})}\delta$ when $f$ is $\beta$-smooth.

**Lemma E.1** *Each iteration of Algorithm 1 can be implemented in $O(md^{\omega-1})$ arithmetic operations plus $O(1)$ calls to the oracle for the value of $f$.*

**Proof:**    We go through each step of Algorithm 1 and add up the number of arithmetic operations and oracle calls for each step:

1. Line 4 samples a $d$-dimensional Gaussian random vector $\xi \sim N(0, I_d)$, which can be performed in $O(d)$ arithmetic operations.

2. At each iteration, Algorithm 1 computes the matrix $H(w)$ at $w = \theta$ (line 5) and $w = z$ (line 9):

$$H(w) = \sum_{j=1}^{m} \frac{a_j a_j^\top}{(a_j^\top w - b_j)^2}.$$

   Computing $H(w)$ at any $w \in \mathbb{R}^d$ can be accomplished in $md^{\omega-1}$ operations as follows:

   Define $C(w)$ to be the $d \times m$ matrix where each column $c_j(w) = \frac{a_j}{a_j^\top w - b_j}$ for all $j \in [m]$. Then

$$H(w) = C(w)C(w)^\top.$$

   Since $C(w)$ is a $d \times m$ matrix the product $C(w)C(w)^\top$ can be computed in $md^{\omega-1}$ arithmetic operations. Thus, Lines 5 and 9 of Algorithm 1 can each be computed in $md^{\omega-1}$ arithmetic operations.

3. Since Lines 6 and 10 compute a sum of two $d \times d$ matrices, Lines 6 and 10 can each be performed in $d^2$ arithmetic operations.

4. Line 7 computes the proposed update

$$z = \theta + \Phi(\theta)^{-\frac{1}{2}}\xi.$$

Computing $\Phi(\theta)^{-\frac{1}{2}}$ can be performed by taking the singular value decomposition of the matrix $\Phi(\theta)$, and then inverting and taking the square root of its eigenvalues. This can be accomplished in $d^\omega$ arithmetic operations since $\Phi(\theta)$ is a $d \times d$ matrix. Once $\Phi(\theta)^{-\frac{1}{2}}$ is computed, the computation $\theta + \Phi(\theta)^{-\frac{1}{2}}\xi$ can be performed in $d^2$ arithmetic operations. Thus Line 7 can be computed in $O(d^\omega) \leq O(md^{\omega-1})$ arithmetic operations.

5. Line 8 requires determining whether $z \in K$. This can be accomplished in $O(md)$ arithmetic operations, by checking whether the inequality $Az \leq b$ is satisfied.

6. Line 11 requires computing the determinant $\det(\Phi(w))$ and $f(w)$ for $w = \theta$ and $w = z$. Computing $\det(\Phi(w))$ can be accomplished by computing the singular value decomposition of $\det(\Phi(w))$, and then taking the product of the resulting singular values to compute the determinant. Since $\Phi(w)$ is a $d \times d$ matrix, computing the singular value decomposition can be done in $d^\omega$ arithmetic operations. Computing $f(w)$ for any $w \in \mathbb{R}^d$ can be accomplished in one call to the oracle for the value of $f$. Thus, Line 11 can be computed in $O(d^\omega) \leq O(md^{\omega-1})$ arithmetic operations, plus two calls to the oracle for the value of $f$.

Therefore, adding up the number of arithmetic operations and oracle calls from all the different steps of Algorithm 1, we get that each iteration of Algorithm 1 can be computed in $O(md^{\omega-1})$ arithmetic operations plus $O(1)$ calls to the oracle for the value of $f$.

∎

### E.2 Bounding the step size

**Definition E.1 (Cross-ratio distance)** *Let $u, v \in \mathrm{Int}(K)$. If $u \neq v$, let $p, q$ be the endpoints of the chord in $K$ which passes through $u$ and $v$ such that the four points lie in the order $p, u, v, q$. Define*

$$\sigma(u, v) := \frac{\|u - v\|_2 \times \|p - q\|_2}{\|p - u\|_2 \times \|v - q\|_2}$$

*if $u \neq v$, and $\sigma(u, v) = 0$ if $u = v$.*

For convenience, we define the cross-ratio distance between any two subsets $S_1, S_2 \subseteq K$ as

$$\sigma(S_1, S_2) = \min\{\sigma(u, v) : u \in S_1, v \in S_2\}.$$

And for any $S \subseteq \mathbb{R}^d$ and any density $\nu : \mathbb{R}^d \to \mathbb{R}$ we define the induced measure:

$$\nu^\star(S) = \int_{z \in S} \nu(z)\mathrm{d}z.$$

**Definition E.2** *For any positive-definite matrix $M \in \mathbb{R}^d \times \mathbb{R}^d$, and any $u \in \mathbb{R}^d$, we define*

$$\|u\|_M := \sqrt{u^\top M u}.$$

**Lemma E.2** *For any $u, v \in \mathrm{Int}(K)$, we have*

$$\sigma(u, v) \geq \frac{1}{\sqrt{2m\alpha^{-1} + \eta^{-1}R^2}}\|u - v\|_{\Phi(u)}.$$

**Proof:** Let $p, q$ be the endpoints of the chord in $K$ which passes through $u$ and $v$ such that the four points lie in the order $p, u, v, q$. Then

$$\sigma^2(u, v) = \left( \frac{\|u - v\|_2 \times \|p - q\|_2}{\|p - u\|_2 \times \|v - q\|_2} \right)^2$$

$$\geq \max \left( \frac{\|u - v\|_2^2}{\|p - u\|_2^2}, \frac{\|u - v\|_2^2}{\|u - q\|_2^2}, \frac{\|u - v\|_2^2}{\|p - q\|_2^2} \right)$$

$$= \max \left( \max_{i \in [m]} \frac{(a_i^\top (u - v))^2}{(a_i^\top u - b_i)^2}, \frac{\|u - v\|_2^2}{\|p - q\|_2^2} \right)$$

$$\geq \frac{1}{2} \max_{i \in [m]} \frac{(a_i^\top (u - v))^2}{(a_i^\top u - b_i)^2} + \frac{1}{2} \frac{\|u - v\|_2^2}{\|p - q\|_2^2}$$

$$\geq \left( \frac{1}{2m} \sum_{i=1}^m \frac{(a_i^\top (u - v))^2}{(a_i^\top u - b_i)^2} \right) + \frac{1}{2} \frac{\|u - v\|_2^2}{R^2}$$

$$= (u - v)^\top \left( \frac{1}{2m\alpha^{-1}} \times \alpha^{-1} \sum_{i=1}^m \frac{(a_i a_i^\top)^2}{(a_i^\top u - b_i)^2} + \frac{1}{2R^2\eta^{-1}} \times \eta^{-1} I_d \right) (u - v)$$

$$\geq \frac{1}{2m\alpha^{-1} + 2\eta^{-1}R^2} (u - v)^\top \Phi(u)(u - v)$$

$$= \frac{1}{2m\alpha^{-1} + 2\eta^{-1}R^2} \|u - v\|_{\Phi(u)}^2.$$

Thus, we have

$$\sigma(u, v) \geq \frac{1}{\sqrt{2m\alpha^{-1} + \eta^{-1}R^2}} \|u - v\|_{\Phi(u)}.$$

$\blacksquare$

**Lemma E.3** *For any $u, v \in \text{Int}(K)$ such that $\|u - v\|_{\Phi(u)} \leq \frac{1}{2\alpha^{1/2}}$ we have that*

$$(1 - \alpha^{1/2} \|u - v\|_{\Phi(u)})^2 \Phi(v) \preceq \Phi(u) \preceq (1 + \alpha^{1/2} \|u - v\|_{\Phi(u)})^2 \Phi(v).$$

**Proof:**

$$\|u - v\|_{\Phi(u)}^2 = \alpha^{-1} \sum_{i=1}^m \frac{(a_i^\top (u - v))^2}{(a_i^\top u - b_i)^2} + \eta^{-1} \|u - v\|^2$$

$$\geq \alpha^{-1} \max_{i \in [m]} \frac{(a_i^\top (u - v))^2}{(a_i^\top u - b_i)^2}.$$

Thus,

$$|(a_i^\top u - b_i) - (a_i^\top v - b_i)| \leq \alpha^{1/2} \|u - v\|_{\Phi(u)} |a_i^\top u - b_i| \qquad \forall i \in [m].$$

Therefore, for all $w \in \mathbb{R}^d$ we have

$$w^\top \left[ (1 - \alpha^{1/2} \|u - v\|_{\Phi(u)})^2 \alpha^{-1} \sum_{i=1}^m \frac{a_i a_i^\top}{(a_i^\top v - b_i)^2} + \eta^{-1} I_d \right] w$$

$$\leq w^\top \left[ \alpha^{-1} \sum_{i=1}^m \frac{a_i a_i^\top}{(a_i^\top u - b_i)^2} + \eta^{-1} I_d \right] w$$

$$\leq w^\top \left[ (1 + \alpha^{1/2} \|u - v\|_{\Phi(u)})^2 \alpha^{-1} \sum_{i=1}^m \frac{a_i a_i^\top}{(a_i^\top v - b_i)^2} + \eta^{-1} I_d \right] w.$$

Thus,

$$(1 - \alpha^{1/2} \|u - v\|_{\Phi(u)})^2 \Phi(v) \preceq \Phi(u) \preceq (1 + \alpha^{1/2} \|u - v\|_{\Phi(u)})^2 \Phi(v).$$

$\blacksquare$

### E.3 Bounding the acceptance probability

**Lemma E.4** *Let $\theta \in \text{Int}(K)$. Then the acceptance ratio of satisfies*

$$\mathbb{P}_{z \sim N(\theta, \Phi^{-1}(\theta))} \left( \frac{\pi(z)\sqrt{\det(\Phi(z))}}{\pi(\theta)\sqrt{\det(\Phi(\theta))}} \times \exp\left( \|z - \theta\|_{\Phi(\theta)}^2 - \|\theta - z\|_{\Phi(z)}^2 \right) \times \mathbb{1}\{z \in K\} \geq \frac{3}{10} \right) \geq \frac{1}{3}.$$

**Proof:**

By (33) of Lemma E.9, we have

$$\mathbb{P}_{z \sim N(\theta, \Phi^{-1}(\theta))} \left( \|z - \theta\|_{\Phi(\theta)}^2 - \|z - \theta\|_{\Phi(z)}^2 \geq -\frac{2}{50} \right) \geq \frac{98}{100}. \tag{7}$$

By Lemmas E.5, E.9, and E.8 we have that

$$\mathbb{P}_{z \sim N(\theta, \Phi^{-1}(\theta))} \left( \frac{\pi(z)\sqrt{\det(\Phi(z))}}{\pi(\theta)\sqrt{\det(\Phi(\theta))}} \times \exp\left( \|z - \theta\|_{\Phi(\theta)}^2 - \|\theta - z\|_{\Phi(z)}^2 \right) \times \mathbb{1}\{z \in K\} \geq \frac{3}{10} \right)$$

$$\geq \mathbb{P}_{z \sim N(\theta, \Phi^{-1}(\theta))} \left( \left\{ \frac{\pi(z)}{\pi(\theta)} \geq \frac{1}{2} \right\} \cap \left\{ \frac{\sqrt{\det(\Phi(z))}}{\sqrt{\det(\Phi(\theta))}} \geq \frac{48}{50} \right\} \right.$$

$$\left. \cap \left\{ \exp\left( \|z - \theta\|_{\Phi(\theta)}^2 - \|\theta - z\|_{\Phi(z)}^2 \right) \geq 0.96 \right\} \cap \{z \in K\} \right)$$

$$= 1 - \mathbb{P}_{z \sim N(\theta, \Phi^{-1}(\theta))} \left( \left\{ \frac{\pi(z)}{\pi(\theta)} \geq \frac{1}{2} \right\}^c \cup \left\{ \frac{\sqrt{\det(\Phi(z))}}{\sqrt{\det(\Phi(\theta))}} \geq \frac{48}{50} \right\}^c \right.$$

$$\left. \cup \left\{ \|z - \theta\|_{\Phi(\theta)}^2 - \|\theta - z\|_{\Phi(z)}^2 \geq -\frac{2}{50} \right\}^c \cup \{z \in K\}^c \right)$$

$$\geq 1 - \mathbb{P}_{z \sim N(\theta, \Phi^{-1}(\theta))} \left( \left\{ \frac{\pi(z)}{\pi(\theta)} \geq \frac{1}{2} \right\}^c \right) - \mathbb{P}_{z \sim N(\theta, \Phi^{-1}(\theta))} \left( \left\{ \frac{\sqrt{\det(\Phi(z))}}{\sqrt{\det(\Phi(\theta))}} \geq \frac{48}{50} \right\}^c \right)$$

$$- \mathbb{P}_{z \sim N(\theta, \Phi^{-1}(\theta))} \left( \|z - \theta\|_{\Phi(\theta)}^2 - \|\theta - z\|_{\Phi(z)}^2 < -\frac{2}{50} \right) - \mathbb{P}_{z \sim N(\theta, \Phi^{-1}(\theta))} \left( \{z \in K\}^c \right)$$

$$\overset{\text{Lemmas } E.5, E.9, E.8, \text{ Eq. } 7}{\geq} 1 - \frac{6}{10} - \frac{2}{100} - \frac{2}{100} - \frac{1}{100}$$

$$\geq \frac{1}{3}.$$

∎

**Lemma E.5** *Let $\theta \in \text{int}(K)$. Then*

$$\mathbb{P}_{z \sim N(\theta, \Phi^{-1}(\theta))} \left( \frac{\pi(z)}{\pi(\theta)} \geq \frac{99}{100} \right) \geq \frac{99}{100}.$$

**Proof:** Since $z \sim N(\theta, \Phi(\theta)^{-1})$, we have that

$$z = \theta + \Phi(\theta)^{-\frac{1}{2}}\xi = \theta + (\alpha^{-1}H(\theta) + \eta^{-1}I_d)^{-\frac{1}{2}}\xi$$

for some $\xi \sim N(0, I_d)$.

Since $\alpha^{-1}H(\theta) + \eta^{-1}I_d \succeq \eta^{-1}I_d$, and $H(\theta)$ and $I_d$ are both positive definite, we have that

$$\eta I_d \succeq (\alpha^{-1}H(\theta) + \eta^{-1}I_d)^{-1}. \tag{8}$$

Thus,

$$\|z - \theta\|_2 = \|(\alpha^{-1}H(\theta) + \eta^{-1}I_d)^{-\frac{1}{2}}\xi\|_2$$

$$= \sqrt{\xi^\top (\alpha^{-1}H(\theta) + \eta^{-1}I_d)^{-1}\xi}$$

$$\overset{\text{Eq. (8)}}{\leq} \sqrt{\xi^\top \eta I_d \xi}$$

$$= \sqrt{\eta}\|\xi\|_2. \tag{9}$$

Now, since $\xi \sim N(0, I_d)$, by the Hanson-wright concentration inequality for the $\chi$-distribution [22, 40], we have that

$$\mathbb{P}(\|\xi\|_2 > t) \leq e^{-\frac{t^2-d}{8}} \qquad \forall t > \sqrt{2d}. \tag{10}$$

Thus, Equations (9) and (10) together imply that

$$\mathbb{P}(\|z - \theta\|_2 > \sqrt{\eta}\sqrt{40d}) \leq e^{-\frac{29d}{8}} < \frac{1}{100}. \tag{11}$$

Now, in the setting where $f$ is $L$-Lipschitz, we have

$$\frac{\pi(z)}{\pi(\theta)} = e^{-(f(z)-f(\theta))} \geq e^{-L\|z-\theta\|_2}.$$

Therefore,

$$
\begin{aligned}
\mathbb{P}_{z \sim N(\theta, \Phi^{-1}(\theta))}\left(\frac{\pi(z)}{\pi(\theta)} \geq \frac{99}{100}\right) &\geq \mathbb{P}_{z \sim N(\theta, \Phi^{-1}(\theta))}\left(e^{-L\|z-\theta\|_2} \geq \frac{99}{100}\right) \\
&= \mathbb{P}_{z \sim N(\theta, \Phi^{-1}(\theta))}\left(\|z - \theta\|_2 \leq \frac{\log(\frac{100}{99})}{L}\right) \\
&\geq \frac{99}{100}
\end{aligned}
$$

where the last inequality holds by Inequality (11), since $\eta \leq \frac{1}{10^4 dL^2}$.

Moreover, in the setting where $f$ is differentiable and $\beta$-Lipschitz, we have that, since $z - \theta$ is a multivariate Gaussian random variable,

$$\mathbb{P}((z - \theta)^\top \nabla f(\theta) \leq 0) = \frac{1}{2}.$$

If $(z - \theta)^\top \nabla f(\theta) \leq 0$, we have that

$$
\begin{aligned}
f(z) - f(\theta) &\leq (z - \theta)^\top \nabla f(\theta) + \beta\|z - \theta\|_2^2 \\
&\leq \beta\|z - \theta\|_2^2.
\end{aligned}
$$

Therefore,

$$
\begin{aligned}
&\mathbb{P}_{z \sim N(\theta, \Phi^{-1}(\theta))}\left(\frac{\pi(z)}{\pi(\theta)} \geq \frac{99}{100}\right) \\
&\geq \mathbb{P}_{z \sim N(\theta, \Phi^{-1}(\theta))}\left(\left\{\frac{\pi(z)}{\pi(\theta)} \geq \frac{99}{100}\right\} \cap \left\{(z - \theta)^\top \nabla f(\theta) \leq 0\right\}\right) \\
&\geq \mathbb{P}_{z \sim N(\theta, \Phi^{-1}(\theta))}\left(e^{-\beta\|z-\theta\|_2^2} \geq \frac{99}{100}\middle| (z - \theta)^\top \nabla f(\theta) \leq 0\right) \times \mathbb{P}((z - \theta)^\top \nabla f(\theta) \leq 0) \\
&= \mathbb{P}_{z \sim N(\theta, \Phi^{-1}(\theta))}\left(\|z - \theta\|_2 \leq \frac{\sqrt{\log(2)}}{\sqrt{\beta}}\middle| (z - \theta)^\top \nabla f(\theta) \leq 0\right) \times \mathbb{P}((z - \theta)^\top \nabla f(\theta) \leq 0) \\
&\geq \frac{99}{100} \times \frac{1}{2},
\end{aligned}
$$

where the last Inequality holds by Inequality (11), since $\eta \leq \frac{1}{10^4 d\beta}$ and since the Gaussian distribution is symmetric about the $d - 1$-dimensional hyperplane $(z - \theta)^\top \nabla f(\theta) = 0$. ∎

**Lemma E.6** *For any $\theta, z \in \mathrm{Int}(K)$ such that $\|\theta - z\|_{\Phi(\theta)} \leq \frac{1}{4\alpha^{1/2}}$, we have that*

$$\|N(\theta, \Phi^{-1}(\theta)) - N(z, \Phi^{-1}(z))\|_{\mathrm{TV}}^2 \leq 3d\alpha\|\theta - z\|_{\Phi(\theta)}^2 + \frac{1}{2}\|\theta - z\|_{\Phi(\theta)}^2$$

The proof of Lemma E.6 is an adaptation of the proof Lemma 3 in [41] to our Markov chain's "soft" barrier function, and follows roughly along the lines of that proof.

**Proof:** Since $\|\theta - z\|_{\Phi(\theta)} \leq \frac{1}{4\alpha^{1/2}}$, by Lemma E.3 we have that

$$(1 - \alpha^{1/2}\|\theta - z\|_{\Phi(\theta)})^2 \Phi(z) \preceq \Phi(\theta) \preceq (1 + \alpha^{1/2}\|\theta - z\|_{\Phi(\theta)})^2 \Phi(z). \qquad (12)$$

From Inequality (12) we have that

$$(1 - \alpha^{1/2}\|\theta - z\|_{\Phi(\theta)})^2 I_d \preceq \Phi(z)^{-\frac{1}{2}}\Phi(\theta)\Phi(z)^{-\frac{1}{2}} \preceq (1 + \alpha^{1/2}\|\theta - z\|_{\Phi(\theta)})^2 I_d.$$

Thus, denoting by $\lambda_i(M)$ the $i$th-largest eigenvalue of any matrix $M \in \mathbb{R}^d \times \mathbb{R}^d$, we have by Inequality (12) that

$$(1 - \alpha^{1/2}\|\theta - z\|_{\Phi(\theta)})^2 \leq \lambda_i(\Phi(z)^{-\frac{1}{2}}\Phi(\theta)\Phi(z)^{-\frac{1}{2}}) \leq (1 + \alpha^{1/2}\|\theta - z\|_{\Phi(\theta)})^2 \qquad \forall i \in [d]. \quad (13)$$

Now the KL-divergence between any to multivariate Gaussian distributions with any means $\mu_1, \mu_2 \in \mathbb{R}^d$ and any covariance matrices $\Sigma_1, \Sigma_2 \in \mathbb{R}^d$ satisfies (see e.g. Section 9 of [10] or Fact 5 in [41])

$$D_{\mathrm{KL}}\left(N(\mu_1, \Sigma), N(\mu_2, \Sigma)\right)$$
$$= \frac{1}{2}\left(\mathrm{Tr}(\Sigma_1^{-1}\Sigma_2) - d + \log\left(\frac{\det(\Sigma_1)}{\det(\Sigma_2)}\right) + (\mu_1 - \mu_2)^\top\Sigma_1^{-1}(\mu_1 - \mu_2)\right). \qquad (14)$$

Therefore we have that

$$\|N(\theta, \Phi^{-1}(\theta)) - N(z, \Phi^{-1}(z))\|_{\mathrm{TV}}^2 \overset{\text{Pinsker's Inequality}}{\leq} 2D_{\mathrm{KL}}\left(N(\theta, \Phi^{-1}(\theta)), N(z, \Phi^{-1}(z))\right)$$

$$\overset{\text{Eq. (14)}}{=} \frac{1}{2}\left(\mathrm{Tr}(\Phi(\theta)\Phi^{-1}(z)) - d + \log\left(\frac{\det(\Phi^{-1}(\theta))}{\det(\Phi^{-1}(z))}\right) + (\theta - z)^\top\Phi(\theta)(\theta - z)\right)$$

$$= \frac{1}{2}\left(\mathrm{Tr}(\Phi^{-\frac{1}{2}}(z)\Phi(\theta)\Phi^{-\frac{1}{2}}(z)) - d + \log\left(\frac{\det(\Phi^{-1}(\theta))}{\det(\Phi^{-1}(z))}\right) + (\theta - z)^\top\Phi(\theta)(\theta - z)\right)$$

$$= \frac{1}{2}\sum_{i=1}^d \lambda_i(\Phi^{-\frac{1}{2}}(z)\Phi(\theta)\Phi^{-\frac{1}{2}}(z)) - \frac{1}{2}d + \frac{1}{2}\log\left(\frac{1}{\det(\Phi^{-\frac{1}{2}}(z)\Phi(\theta)\Phi^{-\frac{1}{2}}(z))}\right) + \frac{1}{2}\|\theta - z\|_{\Phi(\theta)}^2$$

$$= \frac{1}{2}\sum_{i=1}^d \left(\lambda_i(\Phi^{-\frac{1}{2}}(z)\Phi(\theta)\Phi^{-\frac{1}{2}}(z)) - 1 + \log\left(\frac{1}{\lambda_i(\Phi^{-\frac{1}{2}}(z)\Phi(\theta)\Phi^{-\frac{1}{2}}(z))}\right)\right) + \frac{1}{2}\|\theta - z\|_{\Phi(\theta)}^2$$

$$\leq \frac{1}{2}\sum_{i=1}^d \left(\lambda_i(\Phi^{-\frac{1}{2}}(z)\Phi(\theta)\Phi^{-\frac{1}{2}}(z)) + \frac{1}{\lambda_i(\Phi^{-\frac{1}{2}}(z)\Phi(\theta)\Phi^{-\frac{1}{2}}(z))} - 2\right) + \frac{1}{2}\|\theta - z\|_{\Phi(\theta)}^2$$

$$\overset{\text{Eq. (13)}}{\leq} \frac{1}{2}\sum_{i=1}^d \left(\max_{t\in\left[(1-\alpha^{1/2}\|\theta-z\|_{\Phi(\theta)})^2,\ (1+\alpha^{1/2}\|\theta-z\|_{\Phi(\theta)})^2\right]} t + \frac{1}{t} - 2\right) + \frac{1}{2}\|\theta - z\|_{\Phi(\theta)}^2$$

$$= \frac{1}{2}\sum_{i=1}^d \left(\max_{t\in\left[-\alpha^{1/2}\|\theta-z\|_{\Phi(\theta)},\ \alpha^{1/2}\|\theta-z\|_{\Phi(\theta)}\right]} (1+t)^2 + \frac{1}{(1+t)^2} - 2\right) + \frac{1}{2}\|\theta - z\|_{\Phi(\theta)}^2$$

$$\leq \frac{1}{2}\sum_{i=1}^d \left(\max_{t\in\left[-\alpha^{1/2}\|\theta-z\|_{\Phi(\theta)},\ \alpha^{1/2}\|\theta-z\|_{\Phi(\theta)}\right]} 6t^2\right) + \frac{1}{2}\|\theta - z\|_{\Phi(\theta)}^2$$

$$\overset{\text{convexity}}{\leq} \frac{1}{2}\left(\sum_{i=1}^d 6\alpha\|\theta - z\|_{\Phi(\theta)}^2\right) + \frac{1}{2}\|\theta - z\|_{\Phi(\theta)}^2$$

$$= 3d\alpha\|\theta - z\|_{\Phi(\theta)}^2 + \frac{1}{2}\|\theta - z\|_{\Phi(\theta)}^2,$$

where the first inequality is Pinsker's inequality, the second inequality holds because $\log(\frac{1}{t}) \leq \frac{1}{t} - 1$ for all t>0, the fourth inequality holds because $(1 + t)^2 + \frac{1}{(1+t)^2} - 2 \leq 6t^2$ for all $t \in [-\frac{1}{4}, \frac{1}{4}]$, and the fifth inequality holds since $t^2$ is convex for $t \in \mathbb{R}$. $\blacksquare$

For every $j \in \mathbb{N}$, let $m_j = m + \lfloor \alpha \eta^{-1} j^2 \rfloor d$. Consider the matrices $A^j = [A^\top, I_d, \ldots, I_d]^\top$ where $A$ is concatenated with $\frac{m_j - 1}{d}$ copies of the identity matrix $I_d$. And consider the vectors $b^j = (b^\top, j\mathbf{1}^\top, \ldots, j\mathbf{1}^\top)$, where $b$ is concatenated with $\frac{m_j - 1}{d}$ copies of the vector $j\mathbf{1}$, where $\mathbf{1} = (1, \ldots, 1)^\top \in \mathbb{R}^d$ is the all-ones vector. Then $K = \{\theta \in \mathbb{R}^d : A^j \theta \leq b^j\}$, and the Hessian of the corresponding log-barrier functions is

$$H_j(w) := \sum_{i=1}^{m^j} \frac{a_i^j (a_i^j)^\top}{((a_i^j)^\top w - b_i^j)^2}.$$

**Lemma E.7** *For all $w \in \text{int}(K)$ we have that*

$$\lim_{j \to \infty} H_j(w) = \alpha \Phi(w), \tag{15}$$

*uniformly in $w$, and that*

$$\lim_{j \to \infty} (H_j(w))^{-1} = \alpha^{-1} (\Phi(w))^{-1}, \tag{16}$$

*uniformly in $w$. Moreover, for any $\theta \in \text{int}(K)$, any $z \in \frac{1}{2} D_\theta$ and any sequence $\{z_j\}_{j=1}^\infty \subseteq \frac{1}{2} D_\theta$ such that $\lim_{j \to \infty} z_j = z$ uniformly in $z$, we have that*

$$\lim_{j \to \infty} H_j(z_j) = \alpha \Phi(z), \tag{17}$$

*uniformly in $z$, and that*

$$\lim_{j \to \infty} \frac{\det(H_j(z_j))}{\det(H_j(\theta))} = \frac{\det(\Phi(z))}{\det(\Phi(\theta))}, \tag{18}$$

*uniformly in $z$.*

**Proof:**

$$H_j(w) = \sum_{i=1}^{m} \frac{a_i a_i^\top}{(a_i^\top w - b_i)^2} + \lfloor \alpha \eta^{-1} j^2 \rfloor \sum_{i=1}^{d} \frac{e_i e_i^\top}{(e_i^\top w - j)^2}$$

$$= H(w) + \lfloor \alpha \eta^{-1} j^2 \rfloor \sum_{i=1}^{d} \frac{e_i e_i^\top}{(e_i^\top w - j)^2}. \tag{19}$$

Now, since $w \in K \subseteq B(0, R)$, we have that

$$(R - j)^2 \leq (e_i^\top w - j)^2 \leq (-R - j)^2. \tag{20}$$

Thus,

$$\lim_{j \to \infty} \frac{\lfloor \alpha \eta^{-1} j^2 \rfloor}{(e_i^\top w - j)^2} \leq \lim_{j \to \infty} \frac{\lfloor \alpha \eta^{-1} j^2 \rfloor}{(R - j)^2} = \alpha \eta^{-1}, \tag{21}$$

and

$$\lim_{j \to \infty} \frac{\lfloor \alpha \eta^{-1} j^2 \rfloor}{(e_i^\top w - j)^2} \geq \lim_{j \to \infty} \frac{\lfloor \alpha \eta^{-1} j^2 \rfloor}{(-R - j)^2} = \alpha \eta^{-1}. \tag{22}$$

Thus, (19) (21), and (22) together imply that

$$\lim_{j \to \infty} H_j(w) = H(w) + \sum_{i=1}^{d} e_i e_i^\top \times \lim_{j \to \infty} \frac{\lfloor \alpha \eta^{-1} j^2 \rfloor}{(e_i^\top w - j)^2}$$

$$= H(w) + \sum_{i=1}^{d} e_i e_i^\top \times \alpha \eta^{-1}$$

$$= H(w) + \alpha \eta^{-1} I_d$$

$$= \alpha \Phi(w), \tag{23}$$

where Inequalities (21) and (22) imply that the convergence to the limit is uniform in $w$. This proves Equation (15).

Moreover, since $\{z_k\}_{k=1}^\infty \subseteq \frac{1}{2}D_\theta$, and $D_\theta \subseteq K$, we also have that

$$|a_i^\top z_j - b_i| \geq \frac{1}{2}|a_i^\top \theta - b_i|. \tag{24}$$

Therefore,

$$
\begin{aligned}
\lim_{j\to\infty} H_j(z_j) &= H(z_j) + \sum_{i=1}^d e_i e_i^\top \times \lim_{j\to\infty} \frac{\lfloor \alpha\eta^{-1}j^2 \rfloor}{(e_i^\top z_j - j)^2} \\
&= \sum_{i=1}^m \frac{a_i a_i^\top}{(a_i^\top z_j - b_i)^2} + \sum_{i=1}^d e_i e_i^\top \times \lim_{j\to\infty} \frac{\lfloor \alpha\eta^{-1}j^2 \rfloor}{(e_i^\top z_j - j)^2} \\
&= \sum_{i=1}^m \frac{a_i a_i^\top}{(a_i^\top z_j - b_i)^2} + \sum_{i=1}^d e_i e_i^\top \times \alpha\eta^{-1} \\
&\overset{\text{Eq. (24)}}{=} \sum_{i=1}^m \frac{a_i a_i^\top}{(a_i^\top z - b_i)^2} + \sum_{i=1}^d e_i e_i^\top \times \alpha\eta^{-1} \\
&= H(z) + \alpha\eta^{-1} I_d \\
&= \alpha\Phi(z),
\end{aligned}
$$

where the convergence of the limit in the fourth equality holds uniformly in $z$ by (24) and the fact that $\lim_{j\to\infty} z_j = z$. Thus, we have that

$$\lim_{j\to\infty} H_j(z_j) = \alpha\Phi(z) \tag{25}$$

uniformly in $z$. This proves Equation (17).

Moreover, since the determinant is a polynomial in the entries of the matrix, Inequality (23) implies that

$$\lim_{j\to\infty} \det(H_j(w)) = \det(\alpha\Phi(w)), \tag{26}$$

uniformly in $w$.

We also have, plugging Inequality (20) into Equation (19),

$$H(w) + \sum_{i=1}^d e_i e_i^\top \times \frac{\lfloor \alpha\eta^{-1}j^2 \rfloor}{(R-j)^2} \preceq H_j(w) \preceq H(w) + \sum_{i=1}^d e_i e_i^\top \times \frac{\lfloor \alpha\eta^{-1}j^2 \rfloor}{(-R-j)^2}$$

and, hence, that

$$v^\top \left[ H(w) + \frac{\lfloor \alpha\eta^{-1}j^2 \rfloor}{(R-j)^2} I_d \right] v \leq v^\top H_j(w) v \leq v^\top \left[ H(w) + \frac{\lfloor \alpha\eta^{-1}j^2 \rfloor}{(-R-j)^2} I_d \right] v \qquad \forall v \in \mathbb{R}^d. \tag{27}$$

Thus, Inequality (27) implies that

$$H(w) + \frac{\lfloor \alpha\eta^{-1}j^2 \rfloor}{(R-j)^2} I_d \preceq H_j(w) \preceq H(w) + \frac{\lfloor \alpha\eta^{-1}j^2 \rfloor}{(-R-j)^2} I_d \qquad \forall j \in \mathbb{N}. \tag{28}$$

Thus, Inequality (28) implies that

$$\left( H(w) + \frac{\lfloor \alpha\eta^{-1}j^2 \rfloor}{(-R-j)^2} I_d \right)^{-1} \preceq (H_j(w))^{-1} \preceq \left( H(w) + \frac{\lfloor \alpha\eta^{-1}j^2 \rfloor}{(R-j)^2} I_d \right)^{-1} \qquad \forall j \in \mathbb{N}. \tag{29}$$

Thus, since $H(w)$ is positive definite, Inequality (29) together with inequalities (21) and (22) imply that

$$
\begin{aligned}
\lim_{j\to\infty} (H_j(w))^{-1} &= (H(w) + \alpha\eta^{-1} I_d)^{-1} \\
&= \alpha^{-1}\Phi(w)^{-1} \qquad \forall w \in \text{Int}(K)
\end{aligned}
$$

uniformly in $w$. This proves Equation (16).

Moreover, Inequality (28) implies that

$$\det(H_j(w)) \le \det\left(H(w) + \frac{\lfloor \alpha\eta^{-1}j^2 \rfloor}{(R-j)^2}I_d\right)$$

$$\le \left(\lambda_{\max}\left(H(w) + \frac{\lfloor \alpha\eta^{-1}j^2 \rfloor}{(R-j)^2}I_d\right)\right)^d$$

$$\le \left(\lambda_{\max}(H(w)) + \lambda_{\max}\left(\frac{\lfloor \alpha\eta^{-1}j^2 \rfloor}{(R-j)^2}I_d)\right)\right)^d$$

$$\le \left(\lambda_{\max}(H(w)) + 3\alpha\eta^{-1}\right)^d$$

$$= c_1 \qquad \forall j \ge 3R, \tag{30}$$

for some $c_1 > 0$ which does not depend on $j$.

Inequality (27) also implies that

$$\det(H_j(w)) \ge \det\left(H(w) + \frac{\lfloor \alpha\eta^{-1}j^2 \rfloor}{(R-j)^2}I_d\right)$$

$$\ge \left(\lambda_{\min}\left(H(w) + \frac{\lfloor \alpha\eta^{-1}j^2 \rfloor}{(R-j)^2}I_d\right)\right)^d$$

$$\ge \left(\max\left(\lambda_{\min}(H(w)), \ \lambda_{\min}\left(\frac{\lfloor \alpha\eta^{-1}j^2 \rfloor}{(R-j)^2}I_d)\right)\right)\right)^d$$

$$\ge \left(\frac{1}{3}\alpha\eta^{-1}\right)^d$$

$$= c_2 \qquad \forall j \ge 3R, \tag{31}$$

for some $c_2 > 0$ which does not depend on either $j$ or $w$.

Thus, Inequalities (26), (30), and (31) together imply that for any $\theta, z \in \text{int}(K)$ we have that

$$\lim_{j\to\infty} \min\left(\frac{\det(H_j(z))}{\det(H_j(\theta))}, \ 1\right) \overset{\text{Eq. (30),(31)}}{=} \min\left(\frac{\lim_{j\to\infty}\det(H_j(z_j))}{\lim_{j\to\infty}\det(H_j(\theta))}, \ 1\right)$$

$$\overset{\text{Eq. (26),(25)}}{=} \min\left(\frac{\det(\alpha\Phi(z))}{\det(\alpha\Phi(\theta))}, \ 1\right)$$

$$= \min\left(\frac{\det(\Phi(z))}{\det(\Phi(\theta))}, \ 1\right),$$

where the limit converges uniformly in $z$. This proves Equation (18).

$\blacksquare$

**Lemma E.8** *Let $\xi \sim N(0, I_d)$ and let $\theta \in \text{int}(K)$. Then with probability at least $1 - \frac{1}{100}\hat{\delta}$ we have that*

$$\theta + \alpha^{\frac{1}{2}}H^{-1/2}(\theta)\xi \in \frac{1}{2}D_\theta$$

*and*

$$\|\xi\|_2 \le 10\sqrt{d}.$$

**Proof:** Let $z = \theta + \alpha^{\frac{1}{2}}H^{-\frac{1}{2}}(\theta)\xi$. Then since $\xi \sim N(0, I_d)$, by the Hanson-Wright concentration inequality [22, 40], we have that

$$\mathbb{P}(\|\xi\|_2 > t) \le e^{-\frac{t^2-d}{8}} \qquad \forall t > \sqrt{2d}.$$

And hence, since $\alpha^{-1/2}\|z - \theta\|_{H(\theta)} = \|H^{\frac{1}{2}}(\theta)H^{-\frac{1}{2}}(\theta)\xi\|_2 = \|\xi\|_2$, that

$$\mathbb{P}(\|z - \theta\|_{H(\theta)} > \alpha^{1/2}t) \le e^{-\frac{t^2-d}{8}} \qquad \forall t > \sqrt{2d}.$$

Hence,

$$\mathbb{P}(\|z - \theta\|_{H(\theta)} > \alpha^{1/2} 10\sqrt{d}) \leq \frac{1}{100}.$$

Thus, since $\alpha \leq \frac{\log(\frac{1}{\delta})}{100d}$ we have that

$$\mathbb{P}\left(z \in \frac{1}{2}D_\theta\right) = \mathbb{P}\left(\|z - \theta\|_{H(\theta)} \leq \frac{1}{2}\right) \geq 1 - \frac{1}{100}\hat{\delta}.$$

∎

**Lemma E.9** *Consider any $\theta \in \mathrm{int}(K)$, and $\xi \sim N(0, I_d)$. Let $z = \theta + (\Phi(\theta))^{-\frac{1}{2}}\xi$. Then*

$$\mathbb{P}\left(\frac{\det(\Phi(z))}{\det(\Phi(\theta))} \geq \frac{48}{50}\right) \geq 1 - \frac{1}{100}\hat{\delta}, \tag{32}$$

*and*

$$\mathbb{P}\left(\|z - \theta\|^2_{\Phi(z)} - \|z - \theta\|^2_{\Phi(\theta)} \leq \frac{2}{50}\right) \geq \frac{98}{100}. \tag{33}$$

**Proof:** Let $z_j = \theta + \alpha^{\frac{1}{2}} H_j^{-\frac{1}{2}}(\theta)\xi$ for all $j \in \mathbb{N}$. Since $H_j(\theta) \succeq H(\theta)$ for all $j \in \mathbb{N}$, we have that $z_j = \theta + \alpha^{\frac{1}{2}} H_j^{-\frac{1}{2}}(\theta)\xi \in \frac{1}{2}D_\theta$ whenever $\theta + \alpha^{\frac{1}{2}} H(\theta)^{-\frac{1}{2}}\xi \in \frac{1}{2}D_\theta$. Let $E$ be the event that $\|\xi\|_2 \leq 10\sqrt{d}$ and that $\{z_j\}_{j=1}^\infty \subseteq \frac{1}{2}D_\theta$. Thus, by Lemma E.8, we have that

$$\mathbb{P}(E) \geq \frac{99}{100}. \tag{34}$$

Moreover, by Equation (16) of Lemma E.7 we have that $\lim_{j\to\infty} H_j^{-1}(\theta) = \alpha^{-1}\Phi^{-1}(\theta)$, which implies that

$$\begin{aligned}
\lim_{j\to\infty} z_j &= \lim_{j\to\infty} \theta + \alpha^{\frac{1}{2}} H_j^{-\frac{1}{2}}(\theta)\xi \\
&= \theta + \Phi^{-\frac{1}{2}}(\theta)\xi \\
&= z
\end{aligned} \tag{35}$$

uniformly in $\xi$ (and hence uniformly in $z = \theta + \Phi^{-\frac{1}{2}}(\theta)\xi$) whenever the event $E$ occurs. Therefore, by Equation (18) of Lemma E.7 we have that

$$\lim_{j\to\infty} \frac{\det(H_j(z_j))}{\det(H_j(\theta))} = \frac{\det(\Phi(z))}{\det(\Phi(\theta))}, \tag{36}$$

uniformly in $z$ whenever the event $E$ occurs.

Since, for each $j \in \mathbb{N}$, $H_j$ is the Hessian of a log-barrier function for $K$, by last line in the proof of Proposition 6 in [41], we have that, for all $t \in (0, \frac{1}{2}]$ and all $\gamma \in (0, 1]$,

$$\mathbb{P}_{\xi \sim N(0, I_d)}\left(\log\det(H_j(\theta + \frac{\gamma}{\sqrt{d}} H_j(\theta)^{-\frac{1}{2}}\xi)) - \log\det(H_j(\theta)) \geq -\gamma\sqrt{2\log(\frac{1}{t})}\right) \geq 1 - t.$$

Setting $t = \frac{1}{100}\hat{\delta}$, we have

$$\mathbb{P}_{\xi \sim N(0, I_d)}\left(\log\det(H_j(\theta + \alpha^{\frac{1}{2}} H_j(\theta)^{-\frac{1}{2}}\xi)) - \log\det(H_j(\theta)) \geq -\frac{1}{100}\right) \geq 1 - \frac{1}{100}\hat{\delta},$$

since $\alpha \leq \frac{1}{400^2 d \log(\frac{1}{\delta})}$.

Thus,

$$\mathbb{P}_{\xi \sim N(0, I_d)}\left(\frac{\det(H_j(\theta + \alpha^{\frac{1}{2}} H_j(\theta)^{-\frac{1}{2}}\xi))}{\det(H_j(\theta))} \geq \frac{49}{50}\right) \geq 1 - \frac{1}{100}\hat{\delta}. \tag{37}$$

Inequalities (34) and (37) together imply that

$$\mathbb{P}_{\xi \sim N(0, I_d)} \left( \left\{ \frac{\det(H_j(z_j))}{\det(H_j(\theta))} \geq \frac{49}{50} \right\} \cap E \right) \geq 1 - \frac{1}{100} \hat{\delta}. \qquad \forall j \in \mathbb{N}. \tag{38}$$

Moreover, Equation (36) implies that there exists some number $N \in \mathbb{N}$ such that

$$\left\{ \frac{\det(H_N(z_N))}{\det(H_N(\theta))} \geq \frac{49}{50} \right\} \cap E \subseteq \left\{ \frac{\det(\Phi(z))}{\det(\Phi(\theta))} \geq \frac{48}{50} \right\} \cap E. \tag{39}$$

Hence,

$$1 - \frac{1}{100} \hat{\delta} \overset{\text{Eq. (38)}}{\leq} \mathbb{P}_{\xi \sim N(0, I_d)} \left( \left\{ \frac{\det(H_N(z_N))}{\det(H_N(\theta))} \geq \frac{49}{50} \right\} \cap E \right)$$

$$\overset{\text{Eq. (39)}}{\leq} \mathbb{P}_{\xi \sim N(0, I_d)} \left( \left\{ \frac{\det(\Phi(z))}{\det(\Phi(\theta))} \geq \frac{48}{50} \right\} \cap E \right). \tag{40}$$

This proves Inequality (32).

Moreover, since, for each $j \in \mathbb{N}$, $H_j$ is the Hessian of a log-barrier function for $K$, by Proposition 7 in [41], for all $t \in (0, \frac{1}{2}]$, all $\alpha > 0$ such that $\sqrt{\alpha d} \leq \frac{t}{20} \log(\frac{11}{t})^{-\frac{3}{2}}$, and all $j \in \mathbb{N}$, we have that

$$\mathbb{P} \left( \|z_j - \theta\|^2_{H_j(z_j)} - \|z_j - \theta\|^2_{H_j(\theta)} \geq 2t\alpha \right) \leq 1 - t \qquad \forall j \in \mathbb{N}. \tag{41}$$

Thus, Equations (41) and (34) imply that

$$\mathbb{P} \left( \left\{ \|z_j - \theta\|^2_{H_j(z_j)} - \|z_j - \theta\|^2_{H_j(\theta)} \leq \frac{1}{50} \alpha \right\} \cap E \right) \geq \frac{98}{100} \qquad \forall j \in \mathbb{N}, \tag{42}$$

since $\alpha \leq \frac{1}{10^5 d}$.

By Equation (17) of Lemma E.7, Equation (35) implies that

$$\lim_{j \to \infty} H_j(z_j) = \alpha \Phi(z), \tag{43}$$

uniformly in $z$, whenever the event $E$ occurs. Thus, Equation (43) implies that

$$\lim_{j \to \infty} \|z_j - \theta\|^2_{H_j(z_j)} - \|z_j - \theta\|^2_{H_j(\theta)} = \lim_{j \to \infty} (z_j - \theta)^\top H_j(z_j)(z_j - \theta) - (z_j - \theta)^\top H_j(\theta)(z_j - \theta)$$

$$\overset{\text{Eq. (43)}}{=} \lim_{j \to \infty} (z - \theta)^\top \alpha \Phi(z)(z - \theta) - (z - \theta)^\top \alpha \Phi(\theta)(z - \theta)$$

$$= \alpha \|z - \theta\|^2_{\Phi(z)} - \alpha \|z - \theta\|^2_{\Phi(\theta)} \tag{44}$$

uniformly in $z$ (and hence in $\xi = \Phi^{\frac{1}{2}}(\theta)(z - \theta)$) whenever event $E$ occurs. Thus, Equation (44) implies that there exists a number $M \in \mathbb{N}$ such that

$$\left\{ \|z_M - \theta\|^2_{H_M(z_M)} - \|z_M - \theta\|^2_{H_M(\theta)} \leq \frac{1}{50} \alpha \right\} \cap E \subseteq \left\{ \alpha \|z - \theta\|^2_{\Phi(z)} - \alpha \|z - \theta\|^2_{\Phi(\theta)} \leq \frac{2}{50} \alpha \right\} \cap E. \tag{45}$$

Thus,

$$\frac{98}{100} \overset{\text{Eq. (42)}}{\leq} \mathbb{P} \left( \left\{ \|z_M - \theta\|^2_{H_M(z_M)} - \|z_M - \theta\|^2_{H_M(\theta)} \leq \frac{1}{50} \alpha \right\} \cap E \right)$$

$$\overset{\text{Eq. (45)}}{\leq} \mathbb{P} \left( \left\{ \alpha \|z - \theta\|^2_{\Phi(z)} - \alpha \|z - \theta\|^2_{\Phi(\theta)} \leq \frac{2}{50} \alpha \right\} \cap E \right).$$

This proves Inequality (33).

∎

### E.4 Bounding the conductance

**Lemma E.10** *For any $\theta, z \in \text{int}(K)$ we have that*

$$\|P_\theta - P_z\|_{\text{TV}} \leq \frac{29}{30}, \qquad \text{whenever} \qquad \|\theta - z\|_{\Phi(\theta)} \leq 1.$$

**Proof:**  First, we note that,

$$\|P_\theta - P_z\|_{\text{TV}} \leq \|P_\theta - N(\theta, \Phi^{-1}(\theta))\|_{\text{TV}}$$
$$+ \|N(\theta, \Phi^{-1}(\theta)) - N(z, \Phi^{-1}(z))\|_{\text{TV}} + \|P_z - N(z, \Phi^{-1}(z))\|_{\text{TV}}. \quad (46)$$

By Lemma E.6, the middle term on the RHS of (46) satisfies

$$\|N(\theta, \Phi^{-1}(\theta)) - N(z, \Phi^{-1}(z))\|_{\text{TV}} \leq \sqrt{3d\alpha\|\theta - z\|^2_{\Phi(\theta)} + \frac{1}{2}\|\theta - z\|^2_{\Phi(\theta)}} \quad (47)$$

Plugging in our choice of hyperparameter $\alpha = \frac{1}{10^5 d}$, Inequality (47) simplifies to

$$\|N(\theta, \Phi^{-1}(\theta)) - N(z, \Phi^{-1}(z))\|_{\text{TV}} \leq \sqrt{\frac{53}{100}}\|\theta - z\|_{\Phi(\theta)}. \quad (48)$$

Thus, if we can show that $\|P_\theta - N(\theta, \Phi^{-1}(\theta))\|_{\text{TV}} \leq \frac{1}{5}$ for all $\theta \in \text{int}(K)$, we will have that $\|P_\theta - P_z\|_{\text{TV}} \leq \frac{29}{30}$ whenever $\|\theta - z\|_{\Phi(\theta)} \leq \frac{1}{2}$, as desired.

To bound the other two terms on the RHS of (46), we observe that

$$\|P_\theta - N(\theta, \Phi^{-1}(\theta))\|_{\text{TV}} = 1 - \mathbb{E}_{z \sim N(\theta, \Phi^{-1}(\theta))}[q(\theta, z)], \quad (49)$$

where

$$q(\theta, z) := \min\{1, \frac{\pi(z)}{\pi(\theta)} \frac{\sqrt{\det\Phi(z)}}{\sqrt{\det\Phi(\theta)}} \exp(\|z - \theta\|^2_{\Phi(\theta)} - \|\theta - z\|^2_{\Phi(z)}) \times \mathbb{1}\{z \in K\}\}$$

is the acceptance ratio.

By Lemma E.5, we have that $\mathbb{P}_{z \sim N(\theta, \Phi^{-1}(\theta))}\left(\frac{\pi(z)}{\pi(\theta)} \geq \frac{99}{100}\right) \geq \frac{99}{10}$. Therefore,

$$\mathbb{P}_{z \sim N(\theta, \Phi^{-1}(\theta))}[q(\theta, z) \geq \frac{9}{10}]$$

$$\overset{\text{Lemma } E.5}{\geq} 1 - \mathbb{P}_{z \sim N(\theta, \Phi^{-1}(\theta))}\left(\frac{\pi(z)}{\pi(\theta)} < \frac{99}{100}\right) - \mathbb{P}_{z \sim N(\theta, \Phi^{-1}(\theta))}\left(\frac{\sqrt{\det(\Phi(z))}}{\sqrt{\det(\Phi(\theta))}} < \sqrt{\frac{48}{50}}\right)$$

$$- \mathbb{P}_{z \sim N(\theta, \Phi^{-1}(\theta))}\left(e^{\|z-\theta\|^2_{\Phi(\theta)} - \|\theta - z\|^2_{\Phi(z)}} < 0.96\right) - \mathbb{P}_{z \sim N(\theta, \Phi^{-1}(\theta))}(z \notin K)$$

$$\geq 1 - \frac{1}{100} - \frac{2}{100} - \frac{2}{100} - \frac{1}{100}$$

$$\geq \frac{9}{10},$$

where the term with the exponent is bounded by Inequality (33) of Lemma E.9, and the other two terms are bounded by Lemmas E.8 and E.9. Therefore,

$$\|P_\theta - N(\theta, \Phi^{-1}(\theta))\|_{\text{TV}} = 1 - \mathbb{E}_{z \sim N(\theta, \Phi^{-1}(\theta))}[q(\theta, z) \geq \frac{9}{10}] \geq 1 - \frac{9}{10} \times \frac{9}{10} \geq \frac{4}{5}. \quad (50)$$

Plugging Inequalities (48) and (50) into (46), we obtain that $\|P_\theta - P_z\|_{\text{TV}} \leq \frac{29}{30}$ whenever $\|\theta - z\|_{\Phi(\theta)} \leq \frac{1}{2}$, as desired.

∎

We recall the following isoperimetric inequality for a log-concave distribution on a convex body, which uses the cross-ratio distance:

**Lemma E.11 (Isoperimetric inequality for cross-ratio distance (Theorem 2.2 of [32]))**
*Let $\pi : \mathbb{R}^d \to \mathbb{R}$ be a log-concave density, with support on a convex body $K$. Then for any partition of $\mathbb{R}^d$ into measurable sets $S_1, S_2, S_3$, the induced measure $\pi^\star$ satisfies*

$$\pi^\star(S_3) \geq \sigma(S_1, S_2)\pi^\star(S_1)\pi^\star(S_2).$$

For any $\theta \in \text{Int}(K)$, define the random variable $Z_\theta$ to be the step taken by the Markov chain in Algorithm 1 from the point $\theta$. That is, set $z = \theta + \Phi(\theta)^{-\frac{1}{2}}\xi$ where $\xi \sim N(0, I_d)$. If $z \in K$, set $Z_\theta = z$ with probability $\min\left(\frac{\pi(z)\sqrt{\det(\Phi(z))}}{\pi(\theta)\sqrt{\det(\Phi(\theta))}}\exp\left(\|z - \theta\|_{\Phi(\theta)}^2 - \|\theta - z\|_{\Phi(z)}^2\right), 1\right)$. Else, set $z = \theta$.

For any $\theta$, $S \subseteq \mathbb{R}^d$, define the one-step transition probability of the soft-threshold Dikin walk Markov chain to be

$$P_\theta(S) = \mathbb{P}(Z_\theta \in S).$$

The next proposition shows that the soft-threshold Dikin walk Markov chain is reversible and $\pi$ is a stationary distribution of this Markov chain:

**Proposition E.12 (Reversibility and stationary distribution)** *For any $S_1, S_2 \subseteq K$ we have that*

$$\int_{\theta \in S_1} \pi(\theta)P_\theta(S_2)\mathrm{d}\theta = \int_{z \in S_2} \pi(z)P_z(S_1)\mathrm{d}z.$$

**Proof:** Let $\rho_\theta(z) := \frac{\sqrt{\det(\Phi(\theta))}}{(2\text{pi})^{\frac{d}{2}}}e^{-\frac{1}{2}(\theta-z)^\top\Phi(\theta)(\theta-z)}$ for any $\theta, z \in \text{Int}(K)$ be the density of the $N(\theta, \Phi(\theta)^{-1})$ distribution.

$$
\begin{aligned}
\int_{\theta \in S_1} \pi(\theta)P_\theta(S_2)\mathrm{d}\theta &= \int_{\theta \in S_1} \pi(\theta) \int_{z \in S_2} \rho_\theta(z) \times \\
&\qquad \min\left(\frac{\pi(z)\sqrt{\det(\Phi(z))}}{\pi(\theta)\sqrt{\det(\Phi(\theta))}}\exp\left(\|z - \theta\|_{\Phi(\theta)}^2 - \|\theta - z\|_{\Phi(z)}^2\right), 1\right)\mathrm{d}z\mathrm{d}\theta \\
&= \int_{\theta \in S_1} \int_{z \in S_2} \pi(\theta)\rho_\theta(z)\min\left(\frac{\pi(z)\rho_z(\theta)}{\pi(\theta)\rho_\theta(z)}, 1\right)\mathrm{d}z\mathrm{d}\theta \\
&= \int_{z \in S_2} \int_{\theta \in S_1} \pi(\theta)\rho_z(\theta)\min\left(\frac{\pi(\theta)\rho_\theta(z)}{\pi(z)\rho_z(\theta)}, 1\right)\mathrm{d}\theta\mathrm{d}z \\
&= \int_{\theta \in S_1} \pi(\theta)P_\theta(S_2)\mathrm{d}\theta.
\end{aligned}
$$

∎

Define the conductance $\phi$ of the Markov chain to be

$$\phi = \inf_{S \subseteq K : \pi^\star(S) \leq \frac{1}{2}} \frac{1}{\pi^\star(S)} \int_S P_\theta(K\backslash S)\pi(\theta)\mathrm{d}\theta.$$

**Lemma E.13** *The conductance $\phi$ satisfies*

$$\phi \geq \frac{1}{10^4} \frac{1}{\sqrt{2m\alpha^{-1} + \eta^{-1}R^2}}.$$

**Proof:** The proof follows the general format for conductance proofs for geometric Markov chains. Let $S_1 \subseteq K$ and let $S_2 = K\backslash S_1$. Without loss of generality, we may assume that $\pi(S_1) \leq \frac{1}{2}$ (since otherwise we could just swap the names "$S_1$" and "$S_2$"). Let

$$S_1' = \left\{\theta \in S_1 : P_\theta(S_2) \leq \frac{1}{70}\right\},$$

$$S_2' = \left\{z \in S_2 : P_z(S_1) \leq \frac{1}{70}\right\}, \tag{51}$$

and let
$$S_3' = (K \backslash S_1') \backslash S_2'.$$

By Proposition E.12 we have that

$$\int_{\theta \in S_1} \pi(\theta) P_\theta(S_2) d\theta = \int_{\theta \in S_2} \pi(\theta) P_\theta(S_1) d\theta. \tag{52}$$

Moreover, by Lemma E.10, for any $u, v \in \text{Int}(K)$ we have that

$$\|P_u - P_v\|_{\text{TV}} \leq \frac{29}{30} \text{ whenever } \|u - v\|_{\Phi(u)} \leq \frac{1}{2}. \tag{53}$$

Thus, on the one hand, by Lemma E.2, Inequality (53) implies that for any $u, v \in \text{Int}(K)$,

$$\|P_u - P_v\|_{\text{TV}} \leq \frac{29}{30}, \qquad \text{whenever} \qquad \sigma(u, v) \leq \frac{1}{2\sqrt{2m\alpha^{-1} + \eta^{-1}R^2}}. \tag{54}$$

On the other hand, Equations (51) imply that, for any $u \in S_1'$, $v \in S_2'$ we have that

$$\|P_u - P_v\|_{\text{TV}} \geq 1 - P_u(S_2) - P_v(S_1) \geq \frac{68}{70}. \tag{55}$$

Thus, Inequalities (54) and (55) together imply that

$$\sigma(S_1', S_2') > \frac{1}{2\sqrt{2m\alpha^{-1} + \eta^{-1}R^2}}. \tag{56}$$

Moreover by Lemma E.11 we have that

$$\pi^\star(S_3') \geq \sigma(S_1', S_2')\pi^\star(S_1')\pi^\star(S_2'). \tag{57}$$

First, we assume that both $\pi^\star(S_1') \geq \frac{1}{4}\pi^\star(S_1)$ and $\pi^\star(S_2') \geq \frac{1}{4}\pi^\star(S_2)$. In this case we have

$$
\int_{S_1} P_\theta(S_2)\pi(\theta)d\theta \overset{\text{Eq. 52}}{=} \frac{1}{2}\int_{S_1} P_\theta(S_2)\pi(\theta)d\theta + \frac{1}{2}\int_{S_2} P_\theta(S_1)\pi(\theta)d\theta
$$
$$
\overset{\text{Eq. 51}}{\geq} \frac{1}{140}\pi^\star(S_3')
$$
$$
\overset{\text{Eq. 57}}{\geq} \frac{1}{140}\sigma(S_1', S_2')\pi^\star(S_1')\pi^\star(S_2')
$$
$$
\overset{\text{Eq. 56}}{\geq} \frac{1}{280}\frac{1}{\sqrt{2m\alpha^{-1} + \eta^{-1}R^2}}\pi^\star(S_1')\pi^\star(S_2')
$$
$$
\geq \frac{1}{560}\frac{1}{\sqrt{2m\alpha^{-1} + \eta^{-1}R^2}}\min(\pi^\star(S_1'), \ \pi^\star(S_2'))
$$
$$
\geq \frac{1}{2024}\frac{1}{\sqrt{2m\alpha^{-1} + \eta^{-1}R^2}}\min(\pi^\star(S_1), \ \pi^\star(S_2)). \tag{58}
$$

Now suppose that instead either $\pi^\star(S_1') < \frac{1}{4}\pi^\star(S_1)$ or $\pi^\star(S_2') < \frac{1}{4}\pi^\star(S_2)$. If $\pi^\star(S_1') < \frac{1}{4}\pi^\star(S_1)$ then we have

$$
\int_{S_1} P_\theta(S_2)\pi(\theta)d\theta \overset{\text{Eq. 52}}{=} \frac{1}{2}\int_{S_1} P_\theta(S_2)\pi(\theta)d\theta + \frac{1}{2}\int_{S_2} P_\theta(S_1)\pi(\theta)d\theta
$$
$$
\geq \frac{1}{2}\int_{S_1 \backslash S_1'} P_\theta(S_2)\pi(\theta)d\theta
$$
$$
\overset{\text{Eq. 51}}{\geq} \frac{1}{2} \times \frac{3}{4} \times \frac{34}{70}\pi^\star(S_1)
$$
$$
\geq \frac{1}{10}\min(\pi^\star(S_1), \ \pi^\star(S_2)). \tag{59}
$$

Similarly, if $\pi^\star(S_2') < \frac{1}{4}\pi^\star(S_2)$ we have

$$\int_{S_1} P_\theta(S_2)\pi(\theta)\mathrm{d}\theta \overset{\text{Eq. 52}}{=} \frac{1}{2}\int_{S_1} P_\theta(S_2)\pi(\theta)\mathrm{d}\theta + \frac{1}{2}\int_{S_2} P_\theta(S_1)\pi(\theta)\mathrm{d}\theta$$

$$\geq \frac{1}{2}\int_{S_2\setminus S_2'} P_\theta(S_1)\pi(\theta)\mathrm{d}\theta$$

$$\overset{\text{Eq. 51}}{\geq} \frac{1}{2} \times \frac{3}{4} \times \frac{34}{70}\pi^\star(S_2)$$

$$\geq \frac{1}{10}\min(\pi^\star(S_1), \ \pi^\star(S_2)). \tag{60}$$

Therefore, Inequalities (58), (59), and (60) together imply that

$$\frac{1}{\min(\pi^\star(S_1), \ \pi^\star(S_2))}\int_{S_1} P_\theta(S_2)\pi(\theta)\mathrm{d}\theta \geq \frac{1}{10^4}\frac{1}{\sqrt{2m\alpha^{-1}+\eta^{-1}R^2}}. \tag{61}$$

for every partition $S_1 \cup S_2 = K$. Hence, Inequality (61) implies that

$$\phi = \inf_{S\subseteq K:\pi^\star(S)\leq\frac{1}{2}}\frac{1}{\pi^\star(S)}\int_S P_\theta(K\setminus S)\pi(\theta)\mathrm{d}\theta \geq \frac{1}{60}\frac{1}{\sqrt{2m\alpha^{-1}+\eta^{-1}R^2}}.$$

∎

**Definition E.3** *We say that a distribution $\nu$ is w-warm for some $w \geq 1$ with respect to the stationary distribution $\pi$ if $\sup_{z\in K}\frac{\nu(z)}{\pi(z)} \leq w$.*

**Lemma E.14 (Corollary 1.5 of [31])** *Suppose that $\mu_0$ is the initial distribution of a lazy reversible Markov chain with conductance $\phi > 0$ and stationary distribution $\pi$, and let $\mu_i$ be the distribution of this Markov chain after $i \geq 0$ steps. Suppose that $\mu_0$ is w-warm with respect to $\pi$ for some $w \geq 1$. Then for all $i \geq 0$ we have*

$$\|\mu_i - \pi\|_{\mathrm{TV}} \leq \sqrt{w}\left(1 - \frac{\phi^2}{2}\right)^i.$$

**Lemma E.15** *Let $\delta > 0$. Suppose that $f : K \to \mathbb{R}$ is either L-Lipschitz (or has $\beta$-Lipschitz gradient). Suppose that $\theta_0 \sim \mu_0$ where $\mu_0$ is a w-warm distribution with respect to $\pi \propto e^{-f}$ with support on $K$. Let $\mu$ denote the distribution of the output of Algorithm 1 with hyperparameters $\alpha \leq \frac{1}{10^5 d}$ and $\eta \leq \frac{1}{10^4 dL^2}$ (or $\eta \leq \frac{1}{10^4 d\beta}$), if it is run for T iterations. Then if $T \geq 10^9\left(2m\alpha^{-1}+\eta^{-1}R^2\right) \times \log(\frac{w}{\delta})$ we have that*

$$\|\mu - \pi\|_{\mathrm{TV}} \leq \delta.$$

**Proof:** By Lemma E.13 we have that the conductance $\phi$ of the Markov chain in Algorithm 1 satisfies

$$\phi \geq \frac{1}{10^4}\frac{1}{\sqrt{2m\alpha^{-1}+\eta^{-1}R^2}},$$

and hence that

$$T = 10^9\left(2m\alpha^{-1}+\eta^{-1}R^2\right) \times \log(\frac{w}{\delta}) \geq 2\phi^{-2} \times \log(\frac{w}{\delta}).$$

Thus, by Lemma E.13 we have that

$$\|\mu_T - \pi\|_{\mathrm{TV}} \leq \sqrt{w}\left(1 - \frac{\phi^2}{2}\right)^T$$

$$= \sqrt{w}\left(1 - \frac{\phi^2}{2}\right)^{2\phi^{-2}\log(\frac{w}{\delta})}$$

$$\leq \sqrt{w}e^{-\log(\frac{w}{\delta})}$$

$$\leq \delta.$$

∎

**Proof:** [of Theorem 2.1]

**Total variation bound.** Recall that we have set the step size parameters $\alpha = \frac{1}{10^5 d}$ and either $\eta = \frac{1}{10^4 dL^2}$ (if $f$ is $L$-Lipschitz) or $\eta = \frac{1}{10^4 d\beta}$ (if $f$ is $\beta$-smooth). Thus, after running Algorithm 1 for $T = 10^9 \left(2m\alpha^{-1} + \eta^{-1} R^2\right) \times \log(\frac{w}{\delta})$ iterations, by Lemma E.15 we have that the distribution $\mu$ of the output of Algorithm 1 satisfies

$$\|\mu - \pi\|_{\mathrm{TV}} \leq \delta. \tag{62}$$

**Bounding the number of operations.** Moreover, by Lemma E.1 we have that each iteration of Algorithm 1 can be implemented in $O(md^{\omega-1})$ arithmetic operations plus $O(1)$ calls to the oracle for the value of $f$. Thus, the number of iterations $T$ taken by Algorithm 1 is $O((md + dL^2 R^2) \times \log(\frac{w}{\delta}))$ iterations in the setting where $f$ is $L$-Lipschitz and $O((md + d\beta R^2) \times \log(\frac{w}{\delta}))$ iterations in the setting where $f$ is $\beta$-smooth, where each iteration takes one function evaluation and $md^{\omega-1}$ arithmetic operations. ∎

# F  Extension to general barrier functions?

From any point $\theta$, our algorithm proposes a step with Gaussian distribution $N(\theta, \frac{1}{d}(\nabla^2 g(\theta))^{-1})$, where $g$ is the following barrier function

$$g(\theta) = \varphi(\theta) + \hat{\eta}^{-1}\theta^\top \theta, \tag{63}$$

where $\varphi(\theta)$ is a barrier function for the polytope $K$, and the parameter $\hat{\eta}^{-1} = \Omega(\frac{1}{L^2})$ in the setting where $f$ is guaranteed to be $L$-Lipschitz and $\hat{\eta}^{-1} = \Omega(\frac{1}{\beta})$ in the setting where $f$ is guaranteed to be $\beta$-smooth. Thus, most of the probability mass of the Gaussian distribution concentrates in the Dikin ellipsoid $\hat{D}_\theta = \theta + \{w : w^\top (\nabla^2 g(\theta))^{-1} w \leq 1\}$ for the barrier function $g$. For simplicity, in our main result, we assume that $\varphi(\theta) := -\sum_{j=1}^m \log(b_j - a_j^\top \theta)$ (which has self-concordance parameter $m$), however, we can in principle choose any barrier function $\varphi$ for the polytope $K$, such as the entropic barrier [4] or the Lee-Sidford Barrier [29] which have self-concordance parameter roughly $\nu = d$.

To arrive at our barrier function from a more axiomatic approach, we first consider the definition of $\nu$-self concordant barrier function:

**Definition F.1 ($\nu$-self-concordant barrier function for $K$)** *We say that $g$ is a $\nu$-self-concordant barrier function if $g : \mathrm{Int}(K) \to \mathbb{R}$ and $g$ satisfies the following conditions:*

1. ***Convex and differentiable barrier:*** *$g$ is convex and third-order differentiable, and $g(x) \to +\infty$ as $x \to \partial K$.*

2. ***Self-concordance:*** *$\nabla^3 g(x)[h, h, h] \leq 2(\nabla^2 g(x)[h, h])^{3/2}$ for all $h \in \mathbb{R}^d$ (this ensures that the Hessian of the barrier function does not change too much each time the Dikin walk takes a step from $x$ to $z$, that is, $\frac{1}{2}\nabla^2 g(z) \preceq \nabla^2 g(x) \preceq 2\nabla^2 g(z)$)*

3. *$g$ **is $\nu$-self concordant:** $h^\top \nabla g(x) \leq \sqrt{\nu h^\top \nabla^2 g(x) h}$ for every $x \in \mathrm{Int}(K)$, $h \in \mathbb{R}^d$*

The fact that our barrier function (63) satisfies parts (1) and (2) of Definition F.1 follows from the fact that $\varphi$ satisfies Definition F.1 and that $\nabla^3(\theta^\top \theta) = 0$. We discuss the self-concordance parameter $\nu$ for which our barrier function satisfies (63) below.

To ensure that the steps $z \sim N(\theta, (\frac{1}{d}\nabla^2 g(\theta))^{-1})$ proposed by our Dikin walk Markov chain arising from the barrier function $g$ has an $\Omega(1)$ acceptance ratio $e^{f(z)-f(\theta)}$, we require that the function $g$ satisfies the following property. This property says that at least $\frac{1}{4}$ of the volume of the Dikin ellipsoid $\hat{D}_\theta$ is contained in the sublevel set $\{z \in K : f(\theta) < f(\theta) + 2\}$ where the value of $f$ does not increase by more than 2.

**Property F.1** *Dikin ellipsoid mostly contained in sublevel set: At every $\theta \in \mathrm{Int}(K)$, the Dikin Ellipsoid, satisfies*

$$\mathrm{Vol}\left(\hat{D}_\theta \cap \{z \in K : f(z) < f(\theta) + 2\}\right) \geq \frac{1}{4}\mathrm{Vol}(\hat{D}_\theta).$$

When designing a barrier function $g$, there is a trade-off between choosing $g$ such that the self-concordance parameter $\nu$ is small, while at the same time ensuring that Property F.1 holds. Roughly, to make the parameter $\nu$ as small as possible, we would like the Dikin ellipsoid to be large relative to the Hilbert-distance metric for the convex body $K$ (This is because, by Proposition 2.3.2(iii) of [39], any $\nu$-self concordant function $g$ satisfies $(h^\top \nabla^2 g(x)h)^{-\frac{1}{2}} \leq |h|_x \leq (1+3\nu)(h^\top \nabla^2 g(x)h)^{-\frac{1}{2}}$, for any $h \in \mathbb{R}^d$ where $|h|_x := \sup\{\alpha > 0 : x \pm \alpha h \in \{z \in K\}\}$). On the other hand, if we make the Dikin ellipsoid too large (with respect to the sublevel set $\{z \in K : f(z) < f(x) + 2\}$) then Property F.1 will not be satisfied, and the steps proposed by the Dikin walk will have a very low acceptance probability. Setting the hyperparameter $\hat{\eta} = \frac{1}{L^2}$ when $f$ is $L$-Lipschitz or $\hat{\eta} = \frac{1}{\beta}$ when $f$ is $\beta$ smooth ensures that our barrier function in (63) satisfies Property F.1.

In the special case where $f$ is the log-density of the uniform distribution ($f \equiv 0$), or when $f$ is any linear function, we have that $f$ is $\beta$-smooth for $\beta = 0$. Thus, our barrier function (63) we use to encode the geometry of $f$ on $K$ is the same as the barrier function $\varphi$ for the polytope $K$. This is because, since the level sets of linear functions $f$ are half-planes, *any* ellipsoid centered at $\theta$ has at least half of its volume in the sublevel set $\{z \in K : f(z) \leq f(\theta)\} \subseteq \{z \in K : f(z) < f(\theta) + 2\}$, satisfying Property F.1.

The following lemma shows that our barrier function in (63) is $\nu$-self concordant with $\nu = O(\nu' + \hat{\eta} R^2)$, where $\nu'$ is the self-concordance parameters of $\varphi$ (which is $\nu' = m$ if we choose $\varphi$ to be log-barrier function). Thus, our barrier function in (63) is $\nu = O(\nu' + L^2 R^2)$ self-concordant in the setting where $f$ is $L$-Lipschitz, and $\nu = O(\nu' + \beta R^2)$ in the setting where $f$ is $\beta$-smooth:

**Lemma F.2** *Suppose that $\phi(x)$ is a $\nu'$-self concordant barrier function for a convex body $K \subset \mathbb{R}^d$ where $B(0,r) \subseteq K \subseteq B(0,R)$ for some $R > r > 0$. Let $g(x) = \phi(x) + \frac{\alpha}{2} x^\top x$ for some $\alpha > 0$. Then $g$ is $\nu$-self concordant for $\nu = 4\nu' + 4\alpha R^2$.*

The proof of Lemma F.2 is given in Appendix F.1. The polynomial dependence of $\nu$ on $LR$ or $\beta R^2$ is a necessary feature of any objective function satisfying Definition F.1 and Property F.1. In appendix F.2, we give explicit examples of classes of objective functions $f$ and polytopes $K$ for which the minimum value of $\nu$ depends polynomially on $LR$ (and classes of smooth functions $f$ where $\nu$ depends polynomially on $\sqrt{\beta} R$.

An open problem is whether one can design versions of the Dikin walk which sample from log-concave distributions with a runtime that depends only on the dimension $d$, and is independent of $L$, $R$ or $\beta$ and which are invariant to affine transformations. The difficulty in achieving this lies in the fact that (e.g., in the setting where $f$ is $L$-Lipschitz), on the one hand, the level sets of $f$ where most of the probability mass of $\propto e^{-f}$ concentrates may have a diameter roughly $RL$ times smaller than the diameter $R$ of $K$. Thus, to have an $\Omega(1)$ acceptance probability the Dikin walk may need to take steps that are roughly $RL$ times smaller than the diameter of $K$. On the other hand, the isoperimetric inequality currently used to bound the mixing time of the Dikin walk uses a metric– the cross-ratio distance for $K$–which, roughly speaking, defines distances between steps by how quickly these steps approach the boundary of $K$. Thus, measured in the cross-ratio distance, the steps proposed by the Dikin walk are of size roughly proportional to $\frac{1}{RL}$, and mixing time bounds obtained with this isoperimetric inequality thus depend polynomially on $RL$. To obtain mixing time bounds independent of the $R, L, \beta$ one may need to show a new isoperimetric inequality which is based on a different metric that encodes the geometry of all of the level sets of $f$–rather than just the geometry of its support $K$.

## F.1 Proof of Lemma F.2

**Proof:** [Proof of Lemma F.2] For any $x \in \text{int}(K)$, we have

$$\nabla g(x) = \nabla \phi(x) + \alpha x$$

Thus, for any $h \in \mathbb{R}^d$,

$$h^\top \nabla g(x) = h^\top \nabla \phi(x) + \alpha h^\top x \leq h^\top \nabla \phi(x) + \alpha \|h\| \|x\| \leq h^\top \nabla \phi(x) + \alpha R \|h\| \tag{64}$$

Thus,

$$(h^\top \nabla g(x))^2 \overset{\text{Eq. (64)}}{\leq} (h^\top \nabla \phi(x) + \alpha R \|h\|)^2 \tag{65}$$
$$\leq 4(h^\top \nabla \phi(x))^2 + 4(\alpha R \|h\|)^2$$
$$= 4(h^\top \nabla \phi(x))^2 + 4\alpha^2 R^2 h^\top I_d h$$
$$= 4(h^\top \nabla \phi(x))^2 + 4\alpha R^2 h^\top (\alpha I_d) h$$
$$\leq 4\nu' h^\top \nabla^2 \phi(x) h + 4\alpha R^2 h^\top (\alpha I_d) h$$
$$\leq (4\nu' + 4\alpha R^2)(h^\top \nabla^2 \phi(x) h + h^\top (\alpha I_d) h)$$
$$= (4\nu' + 4\alpha R^2)(h^\top (\nabla^2 \phi(x) + \alpha I_d) h)$$
$$= (4\nu' + 4\alpha R^2)(h^\top (\nabla^2 g(x)) h)$$

where the third inequality holds since $\phi$ is $\nu'$ self-concordant. Thus, plugging in $\nu = 4\nu' + 4\alpha R^2$ to (65), we get that

$$h^\top \nabla g(x) \leq \sqrt{\nu(h^\top (\nabla^2 g(x)) h)}.$$

$\blacksquare$

## F.2 Lower bounds for self-concordance parameter

**Lower bounds on $\nu$ for worst-case $L$-Lipschitz $f$.** Consider the $L$-Lipschitz function $f(\theta) = L\|\theta\|_2$ constrained to the convex body $K = \frac{R}{2\sqrt{d}}[-1,1]^d$ which is contained in a ball of radius $R$. In this case, any barrier function $g$ which satisfies Property F.1 has Dikin Ellipsoid $\hat{D}_\theta$ which is contained in the ball $B(0, \frac{8}{L})$. Thus, we have that $\nu \geq \Omega(RL)$. (This is because, by Proposition 2.3.2(iii) of [39], any $\nu$-self concordant function $g$ satisfies $(h^\top \nabla^2 g(\theta) h)^{-\frac{1}{2}} \leq |h|_\theta \leq (1 + 3\nu)(h^\top \nabla^2 g(\theta) h)^{-\frac{1}{2}}$, for any $h \in \mathbb{R}^d$ where $|h|_\theta := \sup\{\alpha > 0 : \theta \pm \alpha h \in \{z \in K\}$. Thus, at $\theta = 0$ and choosing $h = (1, \cdots, 1)$ we have $\frac{|h|_\theta}{\|h\|_2} = R$ and $(h^\top \nabla^2 g(\theta) h)^{-\frac{1}{2}} \leq O(\frac{1}{L})$. Thus, $\nu \geq \Omega\left(\frac{|h|_\theta}{(h^\top \nabla^2 g(\theta) h)^{-\frac{1}{2}}}\right) \geq LR)$.

Since there exists a convex body $K$ for which any self-concordant barrier function satisfying Definition F.1 has self-concordance parameter at least $\nu \geq d$, for any $L, R > 0$ there exists a function $f$ and convex body $K \subset B(0, R)$ such that the self-concordance parameter of every barrier function satisfying both Definition F.1 and Property F.1 is at least $\nu \geq \Omega(\max(d, LR))$.

**Lower bounds on $\nu$ for worst-case $\beta$-smooth $f$.** Consider the $\beta$-smooth function $f(\theta) = \frac{1}{2}\beta\theta^\top \theta$ constrained to the convex body $K = \frac{R}{2\sqrt{d}}[-1,1]^d$. At $\theta = 0$, any ellipsoid $\hat{D}_\theta$ satisfying Property F.1 is contained in the ball $\frac{8}{\sqrt{\beta}}B(0,1)$. Thus, we have that $\nu \geq \frac{R}{\frac{8}{\sqrt{\beta}}}\sqrt{\beta} = \Omega(\sqrt{\beta}R)$ (This is because at $\theta = 0$ we have $\frac{|h|_\theta}{\|h\|_2} = R$ and $(h^\top \nabla^2 g(\theta) h)^{-\frac{1}{2}} \leq O(\frac{1}{\sqrt{\beta}})$ and thus, by Proposition 2.3.2(iii) of [39], we have $\nu \geq \Omega\left(\frac{|h|_\theta}{(h^\top \nabla^2 g(\theta) h)^{-\frac{1}{2}}}\right) \geq \sqrt{\beta}R)$. Thus, for any $\beta, R > 0$ there exists a function $f$ and convex body $K \subset B(0, R)$ such that the self-concordance parameter of every barrier function satisfying both Definition F.1 and Property F.1 is at least $\nu \geq \max(d, \Omega(\beta R))$.

