# OpenReview forum: "Sampling from Structured Log-Concave Distributions via a Soft-Threshold Dikin Walk"
_NeurIPS.cc/2023/Conference — NeurIPS 2023 poster_

### Official Review · Reviewer_D286 · 2023-06-22

**Soundness:** 4 excellent
**Presentation:** 3 good
**Contribution:** 2 fair
**Rating:** 7
**Confidence:** 5

**Summary:**

This paper studies the problem of sampling from a log-concave distribution, parametrized by $\exp(f)$ where $f$ is $L$-Lipschitz, over a $d$-dimensional polytope specified by $m$ halfspaces. Assuming the polytope is contained in a ball of radius $R$, the authors develop an algorithm that generates a sample from a distribution at most $\delta$ TV distance away from the target distribution, in time $O((md+dL^2R^2)\cdot md^{\omega-1}\log(w/\delta))$ where $\omega\approx 2.37$ and the initial point is $w$-warm start. Their approach is an extension from Dikin walk by introducing a soft-threshold regularization term. This result has a wide range of applications in differential private optimization.

It is worth noting that Dikin walk was previously adapted for sampling from a uniform distribution over a polytope, and the iteration count and arithmetic operations per iteration are nearly tight [LLV20]. For sampling from a log-concave distribution, hit-and-run has its runtime depends on the dimension $d$, but under certain regimes, it is slower than the proposed algorithm in this paper.

[LLV20]: Laddha, Lee and Vempala. Strong self-concordance and sampling. STOC'20.

**Strengths:**

Under many regimes, this paper provides the state-of-the-art result for sampling from a log-concave distribution over a polytope. Moreover, their algorithm is conceptually easy to understand and should be easy to implement, as it is a variant of the standard Dikin walk for sampling from a uniform distribution. The soft-threshold regularization term is not completely novel, but prior work gets a highly sub-optimal dependence on the self-concordance parameter $\nu$, at least for log-barrier.

Consequently, this algorithm also leads to many applications, such as differential private empirical risk minimization and Bayesian Lasso logit regression.

**Weaknesses:**

There are several weaknesses of this paper to highlight.

1. Restriction to log-barrier. Note that the $md$ iteration bound comes from the log-barrier is $m$-self-concordance. It is well-known that if $m\gg d$, better barrier functions with better parameters exist, such as the hybrid barrier and Lewis weights barrier. However, if one carefully examines the techniques employed in this paper, they are tuned very specified towards log-barrier. In particular, the authors prove that in the limit, sampling from the regularized Gaussian distribution over a polytope whose constraint contains infinitely many copies of identities will converge to sample from the uniform distribution over another polytope. This enables them to utilize prior rich works on Dikin walk for uniform sampling over polytopes to conclude the proof. Unfortunately, this approach does not scale with more complicated barrier functions. Take the volumetric function as an example, the Hessian itself reweights all constraints based on their leverage scores, and duplicating a constraint infinitely many times will make the leverage score goes to 0, invalidating the construction completely. I think it will be much more interesting if the proposed framework can generalize to other barrier functions.

2. Tightness of arithmetic operations. The algorithm takes $O(md^{\omega-1})$ to form the Hessian matrix. This does not seem to be tight. I think the results in this paper will be much stronger if, at least for log-barrier, the arithmetic operations per iteration are tight (say nearly linear in $md$).

3. Dependence on $L$ and $R$. The Dikin walk is inherently a second-order method, so one would expect a better dependence on $L$ and $R$, hopefully polylog dependence. Notably, the Dikin walk for sampling from uniform distribution over a polytope does not depend on $R$, as the walk exploits the local geometry of the Dikin ellipsoid. Hit-and-run can achieve such a result only if the body is in isotropic position [CE22]. The quadratic dependence on $R$ weakens the result significantly in my opinion. Authors should discuss whether the dependence on $R$ can be improved.

Despite the above concerns, I think this paper is technically solid and has many interesting applications.

[CE22]: Yuansi Chen and Ronen Eldan. Hit-and-run via localization schemes. 2022.

**Questions:**

See weakness. I'm willing to raise score if authors can address my concerns.

**Limitations:**

Yes.

---

> ### Author Rebuttal · Authors · 2023-08-09
>
> Thank you for your valuable comments and suggestions.  We are glad that you appreciate our state-of-the-art results, and its many applications to training differentially private and Bayesian ML models, and we thank you for supporting our paper.  We answer your specific questions below.
>
> **“…generalize to other barrier functions”**
>
>
> It is an interesting question whether one can extend our results to more general barrier functions. In Appendix $F$ we show that, if any $\nu’$-self-concordant barrier function for the polytope $K$ is used in place of the log-barrier function in our algorithm, then our regularized barrier function is self-concordant with parameter $\nu = \nu’ + L^2 R^2$ (in, e.g., the setting where $f$ is $L$-Lipschitz). In the special case where $f$ is constant, [Laddha, Lee, Vempala, STOC 2020] show that their Dikin walk Markov chain has mixing time $O(\bar{\nu} d)$, but they require the barrier function to satisfy a stronger condition,  “strong self-concordance with symmetric self-concordance parameter” $\bar{\nu}$ (in particular, they show that the Lewis weights barrier satisfies this stronger condition with parameter $\bar{\nu}=d$). Showing that our regularized barrier function is *strongly* self-concordant with symmetric self-concordance parameter $\bar{\nu} = d + L^2 R^2$ when, e.g., the Lewis weights barrier is used, is thus a natural direction for future work.
>
>
> **"The algorithm takes $O(m d^{\omega-1})$ to form the Hessian matrix. This does not seem to be tight.”**
>
> Investigating whether one can reduce the cost of computing the log-barrier Hessian matrix at each step of our algorithm is an interesting open problem. In particular, we note that [Laddha, Lee, Vempala, STOC 2020] show, in the special case where $f$ is constant, that the (average) cost of computing the Hessian matrix of the log-barrier of the polytope $K:= $ {$\theta \in \mathbb{R}^d : A \theta \leq b$} at each step of their Dikin walk can be improved to roughly $O(d^2 + \textrm{nnz}(A))$ arithmetic operations, where $\textrm{nnz}(A)$ denotes the number of non-zero entries of $A$. Whether their result can be extended to the problem of computing the regularized barrier functions used in our algorithms, in the more general setting where $f$ is $L$-Lipschitz or $\beta$-smooth, is an interesting direction for future work.  We will discuss this in the Conclusions, limitations, and future work section.
>
> **“Dependence on $L$ and $R$...”**
>
> It may be possible to eliminate the polynomial dependence on $L$ and $R$, but it is outside the scope of the current paper.  This is a challenging problem, and we discuss it in the Conclusions, limitations, and future work section, and in more detail in Appendix $F$.
>
> One challenge in obtaining bounds which are independent of $L$ and $R$ is that the isoperimetric inequality used (in our paper, and in many prior works on the Dikin walk) to bound the mixing time of Dikin walk Markov chains relies on a metric—the cross-ratio distance for the polytope $K$— which, roughly speaking, defines the distances between Markov chain steps by how quickly these steps approach the boundary of the polytope $K$. Measured in the cross-ratio distance, the steps proposed by our Dikin walk (or, more generally, by versions of the Dikin walk which take steps that are small enough such that the term $e^{f(z)-f(\theta)}$ in the Metropolis acceptance probability is $\Omega(1)$ when $f$ is $L$-Lipschitz on a polytope contained in a ball of radius $R$) may be as small as roughly $O(\frac{1}{RL})$ with respect to the cross-ratio distance metric, and the mixing time bounds one would obtain via the aforementioned isoperimetric inequality would depend polynomialy on $RL$.

---

> > ### Comment · Reviewer_D286 · 2023-08-10
> >
> > Thank you for your response! Overall, I think the results and quality of this submission is above the bar of NeurIPS, and I'll raise my score to 7.

---

### Official Review · Reviewer_ACUh · 2023-07-05

**Soundness:** 3 good
**Presentation:** 1 poor
**Contribution:** 2 fair
**Rating:** 7
**Confidence:** 3

**Summary:**

The paper proposes a sampling method for log-concave distribution with bounded polytope. The methods can be used in privacy preservation. The theoretical results show the guarantee of the sampling methods with some accepted error magnitude. The comparisons for other methods are also detailedly proposed, showing the improvement of the proposed one.

**Strengths:**

Pros:
1. The work focuses on an important problem in the privacy field--sampling and also provides some ways for us to adopt their methods.
2. The theoretical results show the effectiveness of the proposed methods in magnitude.
3. Some interesting discussion and delicate application in the appendix is provided for better practical usage.

**Weaknesses:**

Cons:
1. Work in the main context is not friendly to the readers. Maybe authors can use more paras to provide different information. The current version is not so friendly for me to read at least. Some more important Lemma/Corollary can also be organized, I believe. I think the design of the func for the barrier should also be clearly stated at first, but not in the Algorithms.
2. Some technical contributions can also be provided to better differentiate yours and the Dikin walk with simply applying a new regularizer.
3. Some numerical studies can be provided to better show the effectiveness of the methods.

**Questions:**

1. can you provide some insights about how your algorithm can speed up the sampling. I just think that is because compared with the Dikin walk, the designed regularizer supports fast computing, but I am not sure.
2. Also, I can not figure out how much improvement for the dimension $d$ in the real problem. Some small examples can be provides to both show the constant in the magnitude and $d$'s improvement in some cases.
3. I am not sure about the adjustment of the line height is allowed.
4. Maybe we also need to consider the time cost for computation of the Lip/smooth constant for $f$

**Limitations:**

Yes

---

> ### Author Rebuttal · Authors · 2023-08-09
>
> Thank you for your valuable comments and suggestions. We are glad you appreciate our theoretical results and the applications to differential privacy, and thank you for supporting our paper. We are sorry for any difficulty understanding our presentation. We answer your specific questions below.
>
> **“technical contributions can also be provided…”**
>
> Our work makes the following novel technical contributions (discussed in Section 3).
>
> **(1)** We introduce self-concordant barrier functions which simultaneously take into account the geometry of both the constraint polytope $K$ and the Lipschitz or smoothness property of the target log-density $f$.
>
> **(2)** The main technical challenge in bounding the mixing time of our Dikin walk is to prove that the determinantal term in the Metropolis acceptance probability $\frac{\textrm{det}\Phi(z)}{\textrm{det}\Phi (\theta)}$ is $\Omega(1)$ with high probability (w.h.p.), where $\Phi$ is the Hessian of our regularized log-barrier function. In previous works on the Dikin walk, which use the log-barrier without regularizer, the determinantal term can be bounded using the following inequality of [Vaidya, Atkinson, 1993] which holds for the Hessian $H$ of the log-barrier:
> $$(\nabla V(\theta))^\top[H(\theta)]^{-1}\nabla V(\theta)\leq O(d)\qquad\forall\theta\in\textrm{int}(K),$$
>
> where $V(\theta):=\log\textrm{det}H(\theta)$. Unfortunately, this inequality does not hold for every self-concordant barrier function. We prove that it *does* however hold for our regularized barrier functions. To see why, we first show that the Hessian of our regularized barrier function, $\Phi(\theta)=\alpha^{-1}H(\theta)+\eta^{-1}I_d$, can be viewed as the limit of an infinite sequence {$H_j(\theta)$}$_{j=1}^\infty$ of matrices, where each $H_j$ is the Hessian of a log-barrier obtained by representing $K$ by an increasing set of (redundant) inequalities. Roughly, this allows us to show that the above inequality, which holds for any log-barrier, must also hold for our regularized barrier function (if we replace $H(\theta)$ with $\Phi(\theta)$ in the above inequality and $V(\theta)$ definition).
>
> **“…insights about how your algorithm can speed up the sampling…”**
>
> The regularizer in our algorithm speeds up the runtime by allowing the Dikin walk to take larger steps, while still ensuring these steps are accepted w.h.p. by the Metropolis accept/reject rule for $f$. Taking larger steps allows our Dikin walk to converge more quickly to the target distribution $\pi\propto e^{-f}$.
>
> To see why, note that from any point $\theta$, the original Dikin walk proposes updates $z=\theta+y$ where $y$ is normally distributed with covariance matrix $\alpha H(\theta)^{-1}$, where $H$ is the Hessian of, e.g., the log-barrier function for $K$ and $\alpha>0$ is a hyperparameter. If one applies the Dikin walk to sample from a non-constant distribution $\pi\propto e^{-f}$, one needs to ensure the stationary distribution it converges to is equal to $\pi$. This can be done by accepting each proposed step with probability proportional to the Metropolis rule $e^{f(z)-f(\theta)}$ and rejecting it otherwise. However, the acceptance probability may be very low (e.g., if half the eigenvalues of $\alpha H(\theta)^{-1}$ are $>\frac{c}{dL^2}$ for some $c>\Omega(1)$, the acceptance probability may be exponentially small in $c$).
>
> One approach (used in [Narayanan, Rakhlin, JMLR 2017]) to ensuring the acceptance probability is high is to choose a smaller $\alpha$ such that all the eigenvalues of $\alpha H(\theta)^{-1}$ are $\leq\frac{1}{dL^2}$, which ensures the acceptance probability $e^{f(z)-f(\theta)}=\Omega(1)$ w.h.p. Unfortunately, this approach can lead the Dikin walk to propose steps with covariance matrix that has many of its eigenvalues unnecessarily small. This is because some eigenvalues of $H(\theta)^{-1}$ may be much larger than other eigenvalues (e.g., if $K$ is much wider in some directions than in others).
>
> To overcome this, our Dikin walk proposes steps with covariance $(\alpha^{-1}H(\theta)+\eta^{-1}I_d)^{-1}$, where we set $\eta=\frac{1}{dL^2}$ (and set $\alpha$ to the same value $\frac{1}{d}$ used in prior works which apply in the special case where $f$ is constant). This ensures the largest eigenvalues of the covariance matrix are no larger than $\frac{1}{dL^2}$, *without* reducing (by more than a constant factor) the eigenvalues which were already $\leq\frac{1}{dL^2}$.
>
> We will add this discussion to Section 3.
>
> **"…Some small examples…"**
>
> Thank you for the suggestion. We will add one or two concrete examples, in addition to the examples given in Section 2 and in the last two columns of Table 1, to better illustrate the runtime improvement.
>
> **“…computation of the Lip/smooth constant”**
>
> Oftentimes, a bound on the Lipschitz or smoothness constant can be calculated analytically. This includes, e.g., applications to training Bayesian or differentially private logistic regression models (or other generalized linear models such as support vector machines (SVM)). In these applications, $f(\theta)=\sum_{i=1}^n\ell(\theta^\top x_i)$ where $\ell:\mathbb{R}\rightarrow\mathbb{R}$ is a convex loss and {$x_1,…,x_n$} $\subset\mathbb{R}^d$ is a dataset. The loss $\ell$ may be $O(1)$-Lipschitz (e.g., if $\ell$ is the logistic loss $\ell(s)=\log(1+e^{-s})$, or the loss $\ell(s)=\max(0,s)$ used to train SVMs) or may be $O(1)$-smooth (in e.g. logistic regression).
>
> When the Lipschitz or smoothness constant is not known, one can in practice set our algorithm's hyperparameters by hand such that the average acceptance probability is, e.g., $>\frac{1}{2}$. One can then run the Markov chain until an easily-computed heuristic convergence metric (e.g., the autocorrelation time) is lower than some desired value (see, e.g., [Durmus, Moulines, “High-dimensional Bayesian inference…” Bernoulli 2019], who use a similar approach to choose hyperparameters of a different Markov chain).
>
> We will add a remark in the final version.

---

> > ### Comment · Reviewer_ACUh · 2023-08-19
> >
> > Thank you for your response! I believe the authors do very good job and I will raise my score to 7

---

### Official Review · Reviewer_wurv · 2023-07-06

**Soundness:** 4 excellent
**Presentation:** 3 good
**Contribution:** 2 fair
**Rating:** 7
**Confidence:** 3

**Summary:**

This paper studies the problem of sampling from distribution of the form $\pi(\theta) \propto e^{-f(\theta)}$, restricted to a polytope $K$. Here, $f$ is either Lipschitz or smooth convex. To this end, the authors propose to use _Dikin walk_ Markov chain (Kannan and Narayanan, 2012), which was originally proposed as a sampler for the uniform distribution on polytopes. Given that $K = \{\theta : A\theta \leq B\}$ with $A=(a_1,\ldots, a_n), B=(b_1,\ldots,b_n)$, the Dikin walk proposes the update $z = \theta + \sqrt{\alpha H^{-1}(\theta)}\xi$ where $\xi\sim N(0,I_d)$, $H$ is the Hessian matrix of the log-barrier function $\varphi(\theta) = -\sum_{j=1}^m \log(b_j-a_j^{\perp}\theta)$, and $\alpha$ is a hyperparameter. If the proposed update is in the interior of $K$, it is accepted with a certain probability, otherwise it is rejected. To ensure that the acceptance probability is $\Omega(1)$ while allowing the Dikin walk to make sufficiently large steps, the authors propose to add the scaled Hessian $\alpha H(\theta)$ by a regularizing term $\eta^{-1}I_d$, in order to "round up" the set of proposals; then, we can adjust $\alpha$ and $\eta$ so that the set of proposals can be fitted in $K$ with high probability. The authors have done a runtime analysis and show that their method can be run in $\tilde{O}(md^{\omega+1})$ in order to sample with small TV error, where $m$ is the dimension of the polytope and $\omega$ is the matrix multiplication constant. In particular, when $m = O(d)$ (e.g. $A$ is full rank), it can be run in $\tilde{O}(d^{\omega+2})$.

**Strengths:**

- The authors did an excellent job on literature review, with complete runtime analysis of previous methods. The time for warm-start and the evaluation of $f$ are also taken into consideration.
- The authors provide detailed explanation on Dikin walk and its limitations.
- The algorithm seems simple enough to implement with a few lines of code.
- The authors provide some motivations for sampling on polytopes in the area of differential privacy.
- I appreciate the overview of the proof of the sampling guarantee in Section 3.



**Weaknesses:**

- The authors mention that the hyperparameter $\alpha$ alone is insufficient for the Dikin walk to make large steps due to the differences in the geometry of the proposals at each point $\theta$. So I think it is better to add a regularizer that depends on $\theta$ that allows us to remove the hyperparameter $\eta$. Have the authors considered such approach? (In other words, can we somehow make $\eta$ depend on $\theta$?)
- The comparison of the runtimes are nice, but I think some experiments on simulated data would make it more convincing that the proposed method is better than the previous ones.
- The authors might want to also discuss the following paper by Chalkis, Fisikopoulos, Papachristou and Tsigaridas: "Truncated Log-concave Sampling for Convex Bodies with Reflective Hamiltonian Monte Carlo".
- In the introduction, the authors need to be more explicit about the matrix multiplication constant $\omega$. Outside readers might not know that $\omega \leq 2.373$, making it harder to compare the numbers in Table 1.

**Questions:**

- There are some notations that should be introduced in the Notation section. For example, $a_j$ and $b_j$. $\Phi(\theta)$ and $\Phi(z)$ are used at the beginning of Section 3 but I could not find their definitions anywhere before this section.
- Maybe I have missed it, but there should be a mention at the beginning on a lower bound of $R/r$ so that it is easier to make comparisons of the numbers in Table 1.

**Limitations:**

The authors have mentioned a limitation of the proposed method that it can only be applied to Lipschitz or $\beta$-smooth $f$. Also, finding a better regularizing term is also a possible research direction.

---

> ### Author Rebuttal · Authors · 2023-08-09
>
> Thank you for your valuable comments and suggestions. We are glad that you appreciate our proof overview, and thank you for supporting our paper. We answer your specific questions below.
>
> **"regularizer that depends on $\theta$..."**
>
> Thank you for the suggestion.  While there are settings where a regularizer that depends on the position $\theta$ may lead to a faster runtime, this would likely require additional access to the function $f,$ or different assumptions on the structure of $f$, beyond what we assume in our paper.
>
> For the class of functions considered in our paper, we conjecture that an $\ell_2$ regularizer which does not depend on $\theta$ is optimal. This is because we consider the class of functions $f$ which are $L$-Lipschitz or $\beta$-smooth with respect to the $\ell_2$ norm, and our bound on $L$ or $\beta$ does not depend on $\theta$. Moreover, we only have access to the function $f$ through an oracle which returns the value of $f$ at any given point $\theta$, but does not tell us how $f$ changes at nearby points.
>
> We will add a remark about this in the final version.
>
>
> **“The authors might want to also discuss the following paper…”**
>
> Thank you for pointing us to this reference, we will discuss it in the related work section.  In particular, we note that the bounds in [Chalkis, Fisikopoulos, Papachristou, Tsigaridas, ACM Transactions on Mathematical Software, 2023] assume that $f$ is $M$-strongly convex for some $M>0$ (we assume $f$ is convex, but not necessarily $M$-strongly convex for $M>0$), and thus are not directly comparable to our bounds.
>
> **“matrix multiplication constant $\omega$.”**  We will add the value of $\omega$ to the introduction.
>
> **“notations…”** Thank you for pointing this out. We will add these definitions to the notation section.
>
>
> **“lower bound of $R/r$...”**
>
> Any convex body satisfies the lower bound $R/r \geq 1$ ($R/r =1$ for a ball, and one can construct a polytope, which approximates a ball, for which $R/r$ is arbitrarily close to $1$).
>
> Moreover, for any convex body $K$, there always exists a linear transformation $T$ for which $TK$ is contained in a ball of radius $R$ and contains a ball of radius $r$ such that $R/r$ satisfies the upper bound $R/r \leq O(\sqrt{d})$ ($TK$ is referred to as a “well-rounded” convex body).
>
> We will add this information to the table caption.

---

> > ### Comment · Reviewer_wurv · 2023-08-17
> > **Response**
> >
> > Thank you for addressing my concerns. I have raised the score by one.
> >
> > Regarding the experiment, can the authors perform some quick experiment on e.g. high-dimensional truncated Dirichlet distribution (in which case $f$ should be convex on a bounded polytope) and add the results to the supplementary?

---

> > > ### Author Response · Authors · 2023-08-21
> > >
> > > Thank you for the suggestion.  We will add an experiment evaluating the runtime and accuracy of our algorithm (and of prior algorithms) when sampling from a log-concave distribution like the one you suggested, in the final version of the paper.

---

### Official Review · Reviewer_5ZqR · 2023-07-07

**Soundness:** 3 good
**Presentation:** 2 fair
**Contribution:** 3 good
**Rating:** 7
**Confidence:** 4

**Summary:**

The paper studies the question of efficient sampling from a log-concave distribution from a constrained polytope. The main idea is to use the well known Dikin's random walk with a soft threshold. The threshold is important to ensure a far more efficient convergence to the desired distribution. This soft threshold ensures that the acceptance ratio is high while also ensuring that the walk remains inside the polytope.

**Strengths:**

The paper gives a definite improvement over the best-known algorithm in the run time for both Lipschitz and smooth loss function. This has direct implications in making DP algorithm for convex optimization more efficient. Just on the strength of the result of the algorithm, I favor acceptance. I have read the proof and did not find any issue with any of it.

**Weaknesses:**

The only weakness of the paper is that the writing can be improved.

**Questions:**

N/A

---

> ### Author Rebuttal · Authors · 2023-08-09
>
> Thank you for your valuable comments, and for taking the time to review our paper. We are glad that you appreciate the strength of our results and we thank you for supporting our paper.

---

### Decision · Program_Chairs · 2023-09-21

**Decision:**

Accept (poster)

**Comment:**

This work studies the algorithmic problem of sampling from a log-concave distribution over a bounded polytope. The main contribution is a probably fast method using the Dikin random walk with a soft threshold regularizer. The reviewers agreed that this is a technically solid contribution that should be accepted to NeurIPS.